# WORDS IN MOTION: EXTRACTING INTERPRETABLE CONTROL VECTORS FOR MOTION TRANSFORMERS

**Ömer Şahin Taş**[*]    **Royden Wagner**[*]
FZI Research Center for Information Technology
Karlsruhe Institute of Technology

## ABSTRACT

Transformer-based models generate hidden states that are difficult to interpret. In this work, we analyze hidden states and modify them at inference, with a focus on motion forecasting. We use linear probing to analyze whether interpretable features are embedded in hidden states. Our experiments reveal high probing accuracy, indicating latent space regularities with functionally important directions. Building on this, we use the directions between hidden states with opposing features to fit control vectors. At inference, we add our control vectors to hidden states and evaluate their impact on predictions. Remarkably, such modifications preserve the feasibility of predictions. We further refine our control vectors using sparse autoencoders (SAEs). This leads to more linear changes in predictions when scaling control vectors. Our approach enables mechanistic interpretation as well as zero-shot generalization to unseen dataset characteristics with negligible computational overhead.

## 1 INTRODUCTION

Accurately predicting sequential data in an interpretable way is desirable for many real-world applications. However, these two objectives often conflict: methods achieving higher accuracy tend to rely on the increased complexity of their underlying models (Kaplan et al., 2020; Bahri et al., 2024). This, in turn, renders them difficult to interpret in terms of semantically meaningful concepts.

Interpretability methods aim to uncover how models process information, often by analyzing whether learned features align with semantically meaningful concepts. The manifold hypothesis (Bengio et al., 2013) suggests that high-dimensional data often lies on a lower-dimensional manifold. Deep learning models learn representations that approximately capture the manifold's geometry by mapping data into a space where related inputs are closer together (Rifai et al., 2011). Therefore, we focus on the structure of the learned representations to make predictions of the model's behavior.

Deep learning models are trained with loss functions that encourage the clustering of data samples in latent space (Papyan et al., 2020). Together with regularizers that prevent overfitting, clusters become more distinct over the course of training, i.e. *neural collapse* (Galanti et al., 2022; Wu & Papyan, 2024). Following Ben-Shaul et al. (2023), we use linear probes (Alain & Bengio, 2017) to measure neural collapse toward interpretable features in hidden states. High probing accuracy implies separability of features, which suggest functionally important directions in hidden states. Building on the insight that interpretable features are embedded in hidden states, we fit control vectors to the directions between hidden states with opposing features.

To further enhance this approach, we use sparse autoencoders to extract more distinct features from hidden states (Bricken et al., 2023). We evaluate sparse autoencoders with fully-connected, convolutional, and MLPMixer layers; and different activation functions. Our experiments with sparse autoencoders of varying sparse intermediate dimensions show that enforcing sparsity leads to more linear changes in prediction when scaling control vectors.

We apply our method to recent multimodal motion transformers (Nayakanti et al., 2023; Zhang et al., 2023b; Wagner et al., 2024). They process features of past motion sequences (i.e., past positions,

---

[*]Equal contribution. Emails: `tas@fzi.de`, `royden.wagner@kit.edu`.
    Our implementation is available at `https://github.com/kit-mrt/future-motion`.

orientation, acceleration, and speed) and environment context (i.e., map data and traffic light states), and transform them into future motion sequences. Like other transformer models, they rely on learned representations of these features, resulting in hidden states that are difficult to interpret and control. We focus on analyzing interpretable motion features that are physically measurable, such as speed, acceleration, direction, and agent type. By leveraging these features, our approach enables interpretable control over generated forecasts and facilitates zero-shot generalization.

Specifically, in this work:

- We argue that, to fit control vectors, latent space regularities with separable features are necessary. We use linear probing and show that neural collapse toward interpretable features occurs in hidden states of recent motion transformers, indicating a structured latent space.

- We fit control vectors using hidden states with opposing features. By modifying hidden states at inference, we show that control vectors describe functionally important directions. Similar to the vector arithmetic in *word2vec*, we obtain predictions consistent with the current driving environment.

- We use sparse autoencoders to optimize our control vectors. Notably, enforcing sparsity leads to more linear changes in predictions when scaling control vectors. We use linearity measures to compare these results against a Koopman autoencoder and SAEs with various layers and activation functions, including convolutional and MLPMixer layers.

## 2 RELATED WORK

**Concept-based interpretability.** Kim et al. (2018) propose explaining predictions with human-interpretable concepts, rather than relying on sample-based raw features. They first choose a set of examples that represent distinct concepts and measure the influence of high-level concepts on the model's decisions, thereby providing global explanations. In a recent text-to-image diffusion setting, Conceptor (Chefer et al., 2024) decomposes a concept into a weighted combination of interpretable elements.

**Structure of the latent space.** Mikolov et al. (2013) show that consistent regularities naturally emerge from the training process of word embeddings. This phenomenon, commonly referred to as the *word2vec* hypothesis, suggests that learned embeddings capture both semantic and syntactic relationships between words through consistent vector offsets in latent space. While the observed linear offsets naturally fit a flat latent space, non-Euclidean geometric models (e.g., Riemannian manifolds) can better capture structural distortions (Arvanitidis et al., 2018). In those cases, "vector arithmetic" can be seen as an approximation to geodesic operations on a curved latent space.

**Neural collapse.** A recent line of work (Papyan et al., 2020; Galanti et al., 2022; Wu & Papyan, 2024) introduces the term *neural collapse* to describe a desirable learning behavior of deep neural networks for classification.[1] It refers to the phenomenon that learned top-layer representations form semantic clusters, which collapse to their means at the end of training. In addition, the cluster means transform progressively into equidistant vectors when centered around the global mean. Therefore, neural collapse facilitates classification tasks and is considered a desirable learning behavior for both supervised (Papyan et al., 2020) and self-supervised learning (Ben-Shaul et al., 2023).

**Hidden state activations.** Transformers consist of attention blocks, followed by simple feed-forward networks, whose hidden state activations are analyzed for interpretability. Elhage et al. (2022) explore two key hypotheses that describe how these activations capture meaningful structures: the linear representation hypothesis (Pennington et al., 2014) and the superposition hypothesis (Arora et al., 2018). These hypotheses essentially state that the neural networks represent features as directions in their activation space, and that representations can be decomposed into independent features.

**Control vectors**[2] are used for a form of activation steering, where concept-based vectors are added to activations (i.e., hidden states) of transformer models. In natural language processing (Zou et al.,

---

[1]Neural collapse is not to be confused with representation collapse (Hua et al., 2021; Barbero et al., 2024), where learned representations across all classes collapse to redundant or trivial solutions (e.g., zero vectors).
[2]Also referred to as steering vectors (Subramani et al., 2022), style vectors (Konen et al., 2024), or activation addition (Turner et al., 2024).

2023; Subramani et al., 2022; Turner et al., 2024; Heo et al., 2025), control vectors allow targeted adjustments to model outputs by modifying hidden states without the need for fine-tuning or prompt engineering. Control vectors are a set of vectors that capture the difference of hidden states with opposing concepts or features (Rimsky et al., 2023). This approach requires a well-structured latent space, where samples are clustered according to classes or features (e.g., a high degree of neural collapse, see Section 3.2).

**Sparse autoencoders.** A key goal of interpretability research is to decompose models and gain a mechanistic interpretation of how their components function. Sparse autoencoders (SAEs) leverage the linear representation hypothesis and approximate the model's activations with a linear combination of feature directions. By enforcing sparsity in latent space, they separate features into distinct, interpretable representations (Bricken et al., 2023; Cunningham et al., 2024; Gao et al., 2025). Related autoencoders linearize learned representations either by manifold flattening (Psenka et al., 2024) or using Koopman operators (Lusch et al., 2019; Azencot et al., 2020).

Our method differs from prior works in several aspects. We measure neural collapse in multimodal models for motion forecasting (i.e., regression) instead of unimodal image classifiers (Papyan et al., 2020) or language models (Wu & Papyan, 2024). Unlike Conmy & Nanda (2024), we do not manually suppress SAE features in control vectors. Furthermore, we do not use our SAEs during inference (Bricken et al., 2023), but to optimize control vectors beforehand, resulting in negligible computational overhead.

## 3 METHOD

### 3.1 MOTION FEATURE CLASSIFICATION USING NATURAL LANGUAGE

In contrast to natural language, where words naturally carry semantic meaning, motion lacks predefined labels. Therefore, we identify human-interpretable motion features by quantizing them into discrete subclasses as in natural language.

Initially, we classify motion direction using the cumulative sum of differences in yaw angles, assigning it to either `left`, `straight`, or `right`. Additionally, we introduce a `stationary` class for stationary objects, where direction lacks semantic significance. We define further classes for speed, dividing the speed values into four intervals: `high`, `moderate`, `low`, and `backwards`. Lastly, we analyze the change in acceleration by comparing the integral of speed over time to the projected displacement with initial speed. Accordingly, we classify acceleration profiles as either `accelerating`, `decelerating`, or `constant` (see Figure 1a). Our thresholds for motion features are based on insights from Ettinger et al. (2021); Seff et al. (2023). The threshold values are detailed in Appendix A.3.

### 3.2 NEURAL COLLAPSE AS A METRIC OF INTERPRETABILITY

We use neural collapse as a metric of interpretability. Specifically, we focus on interpreting hidden states (i.e., activations or latent representations) and evaluate whether hidden states embed interpretable features. We measure how close abstract hidden states are related to interpretable semantics using linear probing accuracy (Alain & Bengio, 2017).[3] We train linear probes (i.e., linear classifiers detached from the overall gradient computation) on top of hidden states ($H_{i,:}$ in Figure 1). During training, we track their accuracy in classifying our interpretable features on validation sets. Adapted to motion forecasting, we choose the aforementioned motion features as interpretable semantics.

Besides linear probing accuracy, following Chen & He (2021), we use the mean of the standard deviation of the $\ell_2$-normalized embedding to measure representation collapse. Representation collapse refers to an undesirable learning behavior where learned embeddings collapse into redundant or trivial representations (Hua et al., 2021; Barbero et al., 2024). Redundant representations have a standard deviation close to zero. In a way, representing the opposite of neural collapse. As shown in (Chen & He, 2021), rich representations have a standard deviation close to $1/\sqrt{\dim}$, where dim is the hidden dimension.

---

[3]Ben-Shaul et al. (2023) show that linear probing accuracy is consistent with the accuracy of nearest class center classifiers, which are typically used to measure neural collapse.

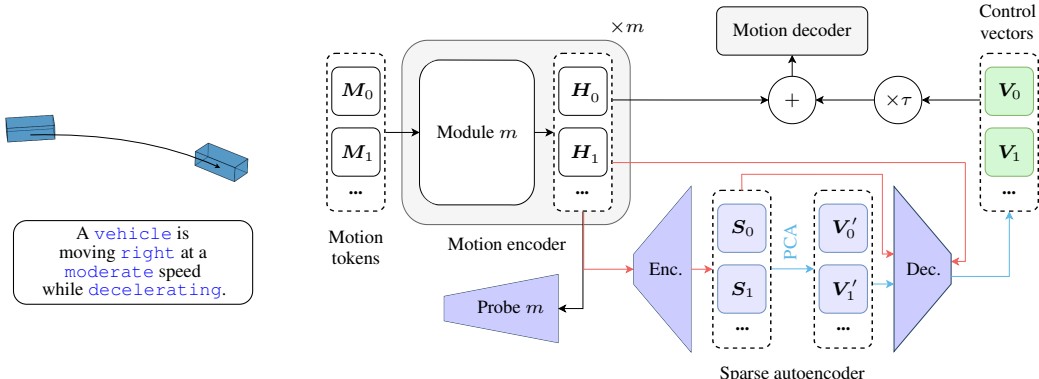

(a) Interpretable motion features          (b) Controllable motion encoder

Figure 1: **Words in Motion.** **(a)** We classify motion features in an interpretable way, as in natural language. **(b)** We measure the degree to which these interpretable features are embedded in the hidden states $\boldsymbol{H}_{i,:}$ of transformer models with linear probes. Furthermore, we use our discrete features and sparse autoencoding to fit interpretable control vectors $\boldsymbol{V}_{i,:}$ that allow for modifying motion forecasts at inference. The training of the sparse autoencoder is shown with red arrows ($\rightarrow$) and the fitting of control vectors with blue arrows ($\rightarrow$).

### 3.3 INTERPRETABLE CONTROL VECTORS

We use our interpretable features to form pairs of opposing features. For each pair, we build a dataset and extract the corresponding hidden states. Next, we compute the element-wise difference between the hidden states of samples with these opposing features. Finally, following Zou et al. (2023) we apply principal component analysis (PCA) with a single component as a pooling method. This reduces the computed differences to a single scalar per hidden dimension to generate control vectors ($\boldsymbol{V}_{i,:}$ in Figure 1b).

We optimize our control vectors using SAEs (Bricken et al., 2023). SAEs extract distinct features in hidden states by encoding and reconstructing them from sparse intermediate representations ($\boldsymbol{S}_{i,:}$ in Figure 1b). We hypothesize that sparse intermediate representations enable a more linear decomposition of our interpretable features, and hence, more distinct control vectors. Therefore, we generate intermediate control vectors $\boldsymbol{V}'_{i,:}$ by pooling the differences between hidden states with opposing features ($\boldsymbol{H}^{\text{pos}}_{i,:}$ vs. $\boldsymbol{H}^{\text{neg}}_{i,:}$). Specifically, we compute

$$\boldsymbol{S}^{\text{pos}}_{i,:} = \text{ReLU}\Big(\boldsymbol{W}_{\text{enc}}\big(\boldsymbol{H}^{\text{pos}}_{i,:} - \boldsymbol{b}_{\text{dec}}\big) + \boldsymbol{b}_{\text{enc}}\Big), \tag{1}$$

where $\boldsymbol{W}$ and $\boldsymbol{b}$ denote weights and biases of the SAE. Similarly, we compute $\boldsymbol{S}^{\text{neg}}_{i,:}$ and obtain the intermediate control vectors as

$$\boldsymbol{V}'_{i,:} = \text{PCA}\big(\boldsymbol{S}^{\text{pos}}_{i,:} - \boldsymbol{S}^{\text{neg}}_{i,:}\big). \tag{2}$$

Leveraging the Johnson-Lindenstrauss Lemma,[4] we use the SAE decoder to project the intermediate control vectors back to the hidden dimension of the motion encoder

$$\boldsymbol{V}_{i,:} = \boldsymbol{W}_{\text{dec}}\boldsymbol{V}'_{i,:} + \boldsymbol{b}_{\text{dec}}. \tag{3}$$

This enables using sparse autoencoders of arbitrary sparse intermediate dimensions for generating control vectors of fixed dimension. At inference, we scale the control vectors with a temperature parameter ($\tau$ in Figure 1b) to control the strength of the corresponding features of a given sample.

---

[4]Johnson & Lindenstrauss (1984) state that a set of points in high-dimensional space can be projected into a lower-dimensional space while approximately preserving the pairwise distances between points.

# 4 EXPERIMENTAL SETUP

## 4.1 MOTION FORECASTING MODELS

We study three recent motion transformers for self-driving. Wayformer (Nayakanti et al., 2023) and RedMotion (Wagner et al., 2024) models employ attention-based scene encoders to learn agent-centric embeddings of past motion, map, and traffic light data. To efficiently process long sequences, Wayformer uses latent query attention (Jaegle et al., 2021) for subsampling, RedMotion lowers memory requirements via local-attention (Beltagy et al., 2020) and redundancy reduction. HPTR (Zhang et al., 2023b) models learn pairwise-relative environment representations via kNN-based attention mechanisms. For Wayformer, we use the implementation by Zhang et al. (2023b) and the early fusion configuration. Therefore, we analyze the hidden states generated by an MLP-based input projector for motion data, which consists of three layers. For RedMotion and HPTR, we use the publicly available implementations. We configure RedMotion with a late fusion encoder for motion data, and HPTR using a custom hierarchical fusion setup with a modality-specific encoder for past motion with a shared encoder for environment context. Further details on model architectures and fusion mechanisms are presented in Appendix A.4 and A.5.

## 4.2 LINEAR PROBES

We add linear probes for our quantized and interpretable motion features (see Section 3.1) to hidden state of all models ($\boldsymbol{H}_{i,:}^{(m)}$ in Figure 1, where $m \in \{0, 1, 2\}$ is the module number and $i$ is the temporal index). These classifiers are learned during training using regular cross-entropy loss to classify speed, acceleration, direction, and the agent classes from hidden states. We decouple this objective from the overall gradient computation. Therefore, these classifiers do not contribute to the alignment of hidden states, but exclusively measure neural collapse toward interpretable features.

## 4.3 CONTROL VECTORS

Using our interpretable motion features, we build pairs of opposing features. Specifically, we generate *speed* control vectors representing the direction from low to high speed, *acceleration* control vectors representing the direction from decelerating to accelerating, and *direction* control vectors representing the direction from turning left to turning right, and *agent* control vectors representing the direction from pedestrian to vehicle. For each pair, we use the hidden states $\boldsymbol{H}_{i,:}^{(m)}$ from module $m = 2$ and the last embedding per motion sequence (with $i = -1$), as it is closest to the start of the prediction.

## 4.4 TRAINING DETAILS AND HYPERPARAMETERS

**Motion transformers.** We provide Wayformer and HPTR models with the nearest 512 map polylines, and RedMotion model with the nearest 128 map polylines. All models process a maximum of 48 surrounding traffic agents as environment context. For the Argoverse 2 Forecasting (abbr. *AV2F*) dataset, we use past motion sequences with 50 time steps (representing 5 s) as input. For the Waymo Open Motion (abbr. *Waymo*) dataset, we use past motion sequences with 11 steps (representing 1.1 s) as input. For Wayformer and RedMotion, we use the unweighted sum of the negative log-likelihood loss for positions modeled as mixture of Gaussians and cross-entropy for confidences as motion forecasting loss. For HPTR, we additionally use the cosine loss for the heading angle and the Huber loss for velocities. We use AdamW (Loshchilov & Hutter, 2019) in its default configuration as optimizer and set the initial learning rate to $2 \times 10^{-4}$. All models have a hidden dimension of 128 and are configured to forecast $k = 6$ trajectories per agent. As post-processing, we follow Konev (2022) and reduce the predicted confidences of redundant forecasts.

**Sparse autoencoders.** We train SAEs as an auxiliary model with sparse intermediate dimensions of 512, 256, 128, 64, 32, and 16. The total loss combines $\ell_2$ reconstruction loss with an $\ell_1$ sparsity penalty: $\ell_2$ ensures accurate reconstruction, while $\ell_1$ promotes sparsity by minimizing small, noise-like activations. The $\ell_1$ must be carefully scaled to avoid deadening important features (Rajamanoharan et al., 2024a). We scale it scaled by $3 \times 10^{-4}$. We optimize the models over $10\,000$ epochs using the Adam optimizer (Kingma & Ba, 2015) and a batch size of 528. The final loss values are provided in Table 7 in the appendix.

## 5    RESULTS

### 5.1    EXTRACTING INTERPRETABLE FEATURES FOR MOTION

Our approach relies on a well-structured latent space, where samples are clustered with respect to interpretable features. First, we ensure that our features are not highly correlated, as confirmed by the Spearman feature correlation analysis in Appendix A.6. Next, we report linear probing accuracy for interpretable features during training on the AV2F and Waymo datasets.

Figure 2 shows the linear probing accuracies for our interpretable motion features for the AV2F dataset. The scores are computed on the validation split over the course of training. All models achieve similar accuracy scores, while the Wayformer model achieves slightly higher scores for classifying acceleration and lower scores for agent classes. Overall, we measure high linear probing accuracy for all intepretable features. This shows that all models likely exhibit neural collapse toward interpretable features.

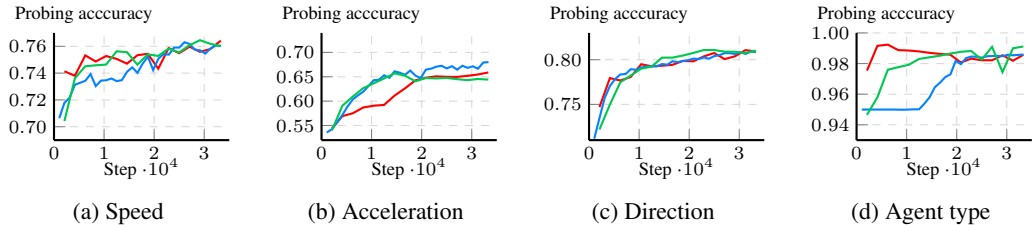

Figure 2: Linear probing accuracies for RedMotion, Wayformer, and HPTR on the validation split of the AV2F dataset.

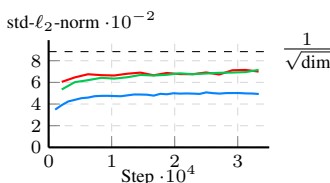

Figure 3: Normalized standard deviation representation quality metric for RedMotion, Wayformer, and HPTR.

The representation quality metric normalized standard deviation of embeddings is shown in Figure 3. Both HPTR and RedMotion learn to generate embeddings with a normalized standard deviation close to the desired value of $1/\sqrt{\text{dim}}$, where dim is the hidden dimension. The scores for Wayformer are lower, which reflects differences between attention-based and MLP-based motion encoders.

Figure 4 shows the linear probing accuracies for our interpretable features on the Waymo dataset. Here, we report the scores for each of the three hidden states $H_i$ in the RedMotion model (i.e., after each module $m$ in the motion encoder, see Figure 1). Similar accuracy scores are reached for all features at all three hidden states. The accuracies for the speed and acceleration classes continuously improve, while those for direction classes reach $0.80$ early on. Compared to the direction scores on the AV2F dataset, the scores on the Waymo dataset "jump" earlier. We hypothesize that this is linked to the shorter input motion sequence on Waymo ($1.1\,\text{s}$ vs. $5\,\text{s}$), which limits the amount possible movements. In contrast to the AV2F dataset, higher accuracies are achieved for classifying speed. Overall, the highest scores are reached for classifying agent types, as on the AV2F dataset.

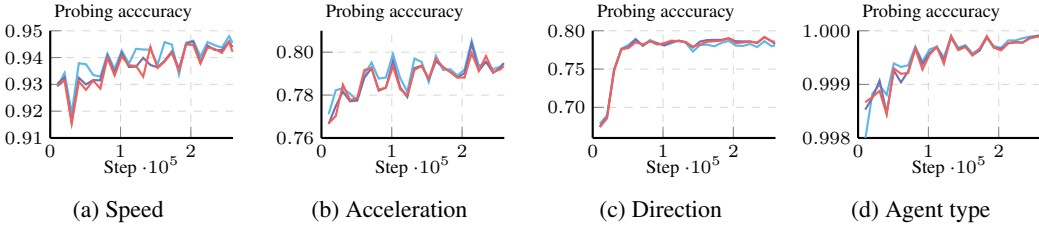

Figure 4: Linear probing accuracies at module 0, module 1 and module 2 for classifiying speed, acceleration, direction, and agent type on the validation split of the Waymo dataset.

In addition to linear probing, we measure neural collapse using class-distance normalized variance (CDNV) (Galanti et al., 2022), see Appendix A.19. On the Waymo dataset, the within-class variance values and the mean distance norm for RedMotion are $10.68$ and $11.24$, respectively, resulting in a CDNV of $0.95$. On the AV2F dataset, these values are $5.73$ and $2.32$, yielding a CDNV of $2.46$. We hypothesize that the higher CDNV value on AV2F is caused by the longer past motion sequence (i.e., $5\,\mathrm{s}$ vs. $1.1\,\mathrm{s}$ on Waymo), allowing for a greater range of potential movements.

## 5.2 MODIFYING HIDDEN STATES OF MOTION TRANSFORMERS AT INFERENCE

Building on the insight that hidden states are likely collapsed toward our interpretable features, we fit control vectors using opposing features. These control vectors allow for modifying motion forecasts at inference. Specifically, we build pairs of opposing features for the AV2F and the Waymo dataset. Then, we fit sets of control vectors ($V_i$ in Figure 1) as described in Section 3.3. At inference, we add the control vectors generated for the last temporal index ($i = -1$) to all embeddings ($i \in \{0, \ldots, 49\}$ for AV2F, $i \in \{0, \ldots, 10\}$ for Waymo).

### 5.2.1 QUALITATIVE RESULTS

Figure 5 shows a qualitative example from the AV2F dataset, where we modify hidden states using our control vector for acceleration scaled with different temperatures $\tau$. Subfigure 5a shows the default (i.e., non-controlled) top-1 (i.e., most likely) motion forecast. In subfigures 5b and 5c, we apply our acceleration control vector with $\tau = -20$ and $\tau = 100$ to enforce a strong deceleration and a moderate acceleration, respectively.

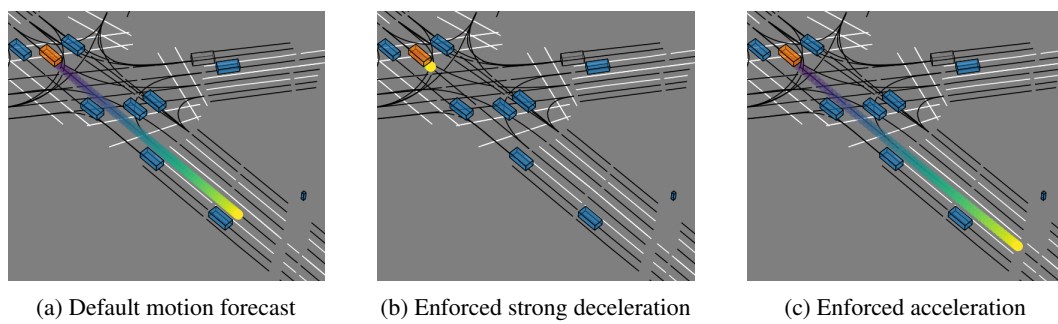

| (a) Default motion forecast | (b) Enforced strong deceleration | (c) Enforced acceleration |

Figure 5: **Modifying hidden states to control a vehicle at an intersection.** We add our acceleration control vector scaled with $\tau = -20$ and $\tau = 100$ to enforce a strong deceleration and a moderate acceleration. The focal agent is highlighted in orange, dynamic agents are blue, and static agents are grey. Lanes are black lines and road markings are white lines.

Figure 6 shows a qualitative example from the Waymo dataset. Subfigure 6a shows the default motion forecast. In subfigures 6b and 6c, we apply our speed control vector to decrease and increase the driven speed of a vehicle. Both modifications affect the future speed in a similar manner, while increasing the speed also changes the route to fit the given environment context (i.e., lanes).

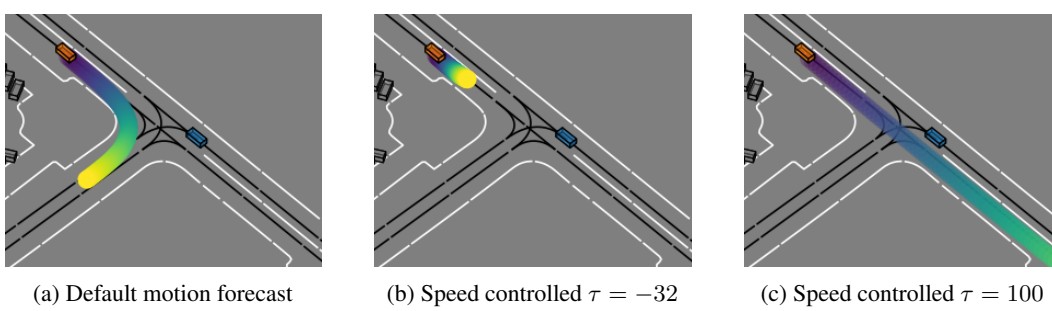

| (a) Default motion forecast | (b) Speed controlled $\tau = -32$ | (c) Speed controlled $\tau = 100$ |

Figure 6: **Modifying hidden states to control a vehicle before a predicted right turn.** In this example, increasing the speed also changes the route to fit the given environment context (i.e., lanes).

In Appendix A.8, we include an example of our direction control vector. Overall, these qualitative results support the finding that the hidden states of motion sequences are arranged with respect to our discrete sets of motion features.

### 5.2.2 SIMILARITY-BASED COMPARISON OF CONTROL VECTORS

In this section, we evaluate how control vectors obtained using SAEs differ from those derived via plain PCA. For comparison, we train SAEs with varying sparse intermediate dimensions: 512, 256, 128, 64, 32, and 16. For each control vector, we calculate its pairwise angles with the control vectors for controlling other features. Table 1 presents the angular distances between control vectors of speed, acceleration, direction, and agent generated with plain PCA and our SAE with a sparse intermediate dimension 128. As expected, the similarity between speed and acceleration, speed and agent, and acceleration and agent is notably high, while the similarity involving direction and other vectors is significantly lower. This result aligns with expectations, as positive speed and acceleration controls lead to faster motion, and our agent control vector represents transition between agent types from pedestrian to vehicle, which is associated with faster motion, as well. Angular-distance results for the remaining SAE dimensions are in Table 6 in the appendix. The similarity of the control vectors generated using the SAE with an intermediate dimension of 128 is the highest.

Table 1: Comparison of control vectors, with angles measured in degrees.

| Plain PCA & Plain PCA | speed | acceleration | direction | agent | SAE & SAE | speed | acceleration | direction | agent |
|---|---|---|---|---|---|---|---|---|---|
| speed | 0.0 | 11.5 | 122.6 | 10.9 | speed | 0.0 | 9.5 | 120.6 | 7.8 |
| acceleration | | 0.0 | 126.8 | 6.8 | acceleration | | 0.0 | 122.9 | 7.0 |
| direction | | | 0.0 | 128.7 | direction | | | 0.0 | 125.8 |
| agent | | | | 0.0 | agent | | | | 0.0 |

### 5.3 QUANTITATIVE EVALUATION OF SAES FOR OPTIMIZING CONTROL VECTORS

We empirically analyze the temporal-causal relationship between modifications on hidden states of past motion and motion forecasts. Specifically, we measure the linearity of relative speed changes in forecasts when scaling our speed control vectors. We use the Pearson correlation coefficient, the coefficient of determination ($R^2$), and the straightness index (S-idx) (Benhamou, 2004) as linearity measures. Given the large range of scenarios in the Waymo dataset, we focus on relative speed changes within a range of $\pm 50\%$ (see Appendix A.14). Higher linearity implies improved controllability.

We compute linearity measures for control vectors optimized using regular SAEs (Bricken et al., 2023) with varying sparse intermediate dimensions. We achieve the highest scores using the SAE with a dimension of 128 (see Table 2). Therefore, we use this dimension in the rest of our evaluations.

Table 2: Scaling sparse autoencoders.

| Autoencoder | Pearson | $R^2$ | S-idx |
|---|---|---|---|
| SAE-512 | 0.990 | 0.974 | 0.984 |
| SAE-256 | 0.990 | 0.974 | 0.985 |
| SAE-128 | **0.993** | **0.984** | **0.988** |
| SAE-64 | 0.991 | 0.976 | 0.985 |
| SAE-32 | 0.990 | 0.959 | 0.985 |
| SAE-16 | 0.982 | 0.770 | 0.958 |

In the following, we evaluate autoencoders with different activation functions and layer types. Following Rajamanoharan et al. (2024b), we use JumpReLU with a threshold $\theta = 0.001$ and regular ReLU activation functions. Moreover, we evaluate regular SAEs with fully-connected layers, with MLPMixer (Tolstikhin et al., 2021) layers (Sparse MLPMixer), and with convolutional layers (ConvSAE). For Sparse MLPMixer and ConvSAE, we use large patch and kernel sizes to approximate the global receptive fields of fully-connected hidden units in regular SAEs. Furthermore, we evaluate a consistent Koopman autoencoder (KoopmanAE) (Azencot et al., 2020) to include a method that models temporal dynamics between embeddings (see Appendix A.15).

Table 3 presents linearity measures for different control vectors derived from both plain PCA pooling and SAE methods. Overall, the regular SAEs (Bricken et al., 2023) achieve the highest Pearson and $R^2$ scores. JumpReLU activation functions improve the $R^2$ scores marginally compared to ReLU activation functions. The SAE version of Cunningham et al. (2024) does not improve the linearity scores. We hypothesize that this is due to reduced decoding flexibility since they transpose the encoder weights instead of learning the decoder weights (i.e., $W_{\text{dec}} = W_{\text{enc}}^{\top}$).

The ConvSAE with a kernel size $k = 64$ and the KoopmanAE achieve the highest straightness index, yet the lowest $R^2$ scores. As shown in Figure 7 and Figure 16 in the appendix, the range of temperatures $\tau$ is much higher for this ConvSAE and significantly lower for the KoopmanAE than for e.g. the regular SAE. This lowers the $R^2$ score but does not affect the straightness index. For the ConvSAE, we hypothesize that this is due to strong activation shrinkage (Rajamanoharan et al., 2024b). Therefore, the JumpReLU configuration of this SAE-type leads to a significantly smaller $\tau$ range (see Appendix A.17), which in turn leads to higher $R^2$ scores (see Table 3). For the KoopmanAE, the opposite is likely, since activation shrinkage is caused by sparsity terms, which are not included in the loss function of Azencot et al. (2020). Notably, activation steering with our SAE-based control vector has an almost 1-to-1 ratio between $\tau$ and relative speed changes (i.e., $\tau = -50$ corresponds to roughly $-50\%$). This improves $R^2$ scores and enables an intuitive interface. Furthermore, improved controllability with SAEs indicates that sparse intermediate representations capture more distinct features.

Table 3: Linearity measures for optimized control vectors: Pearson correlation coefficient, coefficient of determination ($R^2$), and straightness index (S-idx).

| Autoencoder | Activation function | Pooling | Patch/kernel size | Pearson | $R^2$ | S-idx |
|---|---|---|---|---|---|---|
| – | – | PCA | – | 0.988 | 0.969 | 0.981 |
| SAE (Bricken et al., 2023) | ReLU | PCA | – | **0.993** | 0.984 | 0.988 |
| SAE (Rajamanoharan et al., 2024b) | JumpReLU | PCA | – | **0.993** | **0.986** | 0.988 |
| SAE (Cunningham et al., 2024) | ReLU | PCA | – | 0.987 | 0.971 | 0.980 |
| Sparse MLPMixer | ReLU | PCA | 64 | 0.992 | 0.980 | 0.986 |
| Sparse MLPMixer | JumpReLU | PCA | 64 | 0.992 | 0.981 | 0.986 |
| Sparse MLPMixer | ReLU | PCA | 32 | 0.990 | 0.978 | 0.985 |
| Sparse MLPMixer | JumpReLU | PCA | 32 | 0.991 | 0.980 | 0.986 |
| ConvSAE | ReLU | PCA | 64 | 0.986 | 0.383 | 0.991 |
| ConvSAE | JumpReLU | PCA | 64 | 0.987 | 0.861 | 0.978 |
| ConvSAE | ReLU | PCA | 32 | 0.988 | 0.622 | 0.986 |
| ConvSAE | JumpReLU | PCA | 32 | 0.989 | 0.623 | 0.986 |
| KoopmanAE (Azencot et al., 2020) | tanh | PCA | – | 0.991 | $-0.057$ | **1.000** |

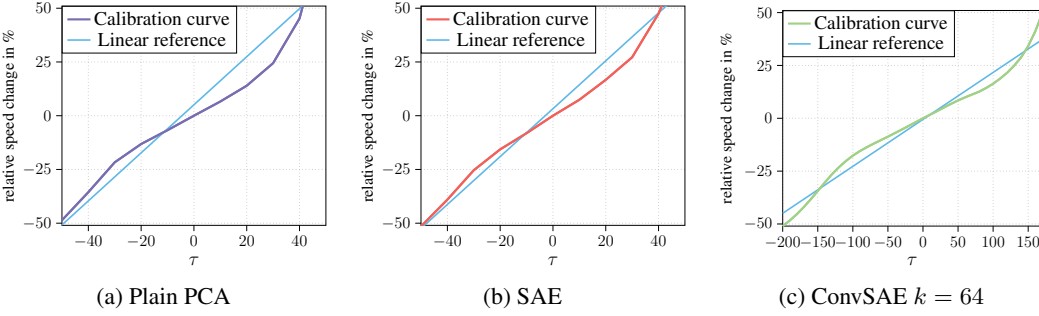

(a) Plain PCA        (b) SAE        (c) ConvSAE $k = 64$

Figure 7: Calibration curves of plain PCA-based speed control vectors and control vectors optimized using SAEs for relative speed changes in forecasts of $\pm 50\%$.

In the appendix, we present an ablation study analyzing our method's sensitivity to hidden states from different modules (see Table 9) and to varying speed thresholds (see Table 10). Our method performs best with a sparse intermediate dimension of $128$ and hidden states from module $m = 2$; and is more sensitive to low than to high speed thresholds.

## 5.4 RELATION OF PROBING ACCURACY TO LINEARITY MEASURES FOR CONTROL VECTORS

We train a RedMotion model on the AV2F dataset using the same trajectory lengths as in the Waymo dataset (1.1 s past and 8 s future), while leaving all other hyperparameters as described in Section 4.4. Table 4 shows the probing accuracy and linearity measures of a speed control vector for this model (see Appendix A.16 for the calibration curve). Compared with a model trained on the Waymo dataset, the AV2F model achieves both a lower probing accuracy and significantly lower linearity measures. These results support our argument that latent space regularities with separable features are necessary to fit precise control vectors.

Table 4: **Higher probing accuracy enables higher linearity measures.** We train RedMotion models on the Waymo and AV2F datasets using the same trajectory lengths. We report the probing accuracies for speed classes and the linearity measures for the corresponding PCA-based control vectors.

| Dataset | Probing accuracy | Pearson | $R^2$ | S-idx |
|---------|------------------|---------|-------|-------|
| AV2F    | 0.753            | 0.877   | 0.275 | 0.891 |
| Waymo   | 0.945            | 0.988   | 0.969 | 0.981 |

## 5.5 ZERO-SHOT GENERALIZATION WITH CONTROL VECTORS

Domain shifts between training and test data significantly degrade the performance of many learning algorithms. Zero-shot generalization methods compensate for such domain shifts without further training or fine-tuning (Kodirov et al., 2015; Xian et al., 2017; Mistretta et al., 2024). In motion forecasting, common domain shifts are more or less aggressive driving styles that result in higher or lower future speeds, respectively. We simulate this domain shift by reducing the future speed in the Waymo validation split by approximately $50\%$. Specifically, we take the first half of future waypoints and linearly upsample this sequence to the original length.

Table 5 shows the results of a RedMotion model trained on the regular training split on this validation split with domain shift. We provide an overview of the used motion forecasting metrics in Appendix A.18. Without the use of our control vectors, high distance-based errors, miss, and overlap rates are obtained. Using the calibration curve in Figure 7b, we compensate for this domain shift by applying our SAE-128 control vector with a temperature $\tau = -50$. This significantly reduces the distance-based errors, the overlap, and the miss rates without further training. In addition, we show the results of applying our control vector with a temperature of $\tau = -30$ and $\tau = -70$, which improves all scores over the baseline as well.

Table 5: Zero-shot generalization to a Waymo dataset version with reduced future speeds. Best scores are **bold**, second best are underlined.

| Control vector | Temperature $\tau$ | minADE$\downarrow$ | Brier minADE$\downarrow$ | minFDE$\downarrow$ | Brier minFDE$\downarrow$ | Overlap rate$\downarrow$ | Miss rate$\downarrow$ |
|----------------|--------------------|--------------------|--------------------------|--------------------|--------------------------|--------------------------|-----------------------|
| None           |                    | 3.271              | 6.547                    | 4.617              | 8.933                    | 0.220                    | 0.580                 |
| SAE-128        | $-30$              | 1.685              | 4.838                    | 2.870              | 8.429                    | 0.179                    | **0.224**             |
| SAE-128        | $-50$              | **1.174**          | **2.759**                | **1.798**          | **4.329**                | **0.174**                | 0.236                 |
| SAE-128        | $-70$              | 1.808              | 3.576                    | 2.035              | 3.676                    | 0.189                    | 0.302                 |

## 6 CONCLUSION

In this work, we take a step toward mechanistic interpretability and controllability of motion transformers. We analyze "words in motion" by examining the representations associated with quantized motion features. Specifically, we show that neural collapse toward interpretable classes of features occurs in recent motion transformers. The high degree of neural collapse indicates a well-separated latent space, that enables to fit precise control vectors to opposing features and modify predictions at inference. We further refine this approach by optimizing our control vectors using sparse autoencoding, resulting in higher linearity. Finally, we compensate for domain shift and enable zero-shot generalization to unseen dataset characteristics. Our findings highlight the effectiveness of sparse dictionary learning and the use of SAEs for improving interpretability.

We assumed a flat latent space and relied on vector arithmetic. We leave a detailed investigation of how SAE parameterization might help address potential latent space curvature for future work. Furthermore, we have empirically shown a connection between neural collapse and the structure of the latent space for using control vectors, although our analysis remained limited to probing accuracy and class-distance normalized variance. Our findings enable new applications in robotics and self-driving. We identify safety validation in latent space as a promising direction, particularly for end-to-end driving. Using control vectors to modify internal representations and adjust trajectories via instruction-based inputs is also a valuable application. Finally, future work can explore the use of other embedding methods (e.g., Schneider et al. (2023)), as well as incorporate features from other modalities by capturing both static and dynamic scene elements.

ACKNOWLEDGMENTS

The research leading to these results is funded by the German Federal Ministry for Economic Affairs and Climate Action within the project "NXT GEN AI METHODS". The authors thank anonymous reviewers for their valuable feedback and insightful suggestions.

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

# A APPENDIX

## A.1 TITLE ORIGIN

The title of our work "words in motion" is inspired by our quantization method using natural language and by a common notion in the computer architecture. In computer architecture, a word is a basic unit of data for a processing unit (e.g., CPU or GPU). In our work, words are classes of motion features that are embedded in the hidden states of motion sequences processed by motion transformers.

## A.2 NATURAL LANGUAGE AS AN INTERFACE FOR MODEL INTERACTION

Linking learned representations to natural language and using it as an interface for model interaction has gained significant attention (e.g., Radford et al. (2021); Alayrac et al. (2022); Liu et al. (2024)). Broadly, approaches incorporating language in models can be categorized into four types. We present these approaches along with applications in robotics and self-driving below.

**Conditioning.** Numerous works use natural language to condition generative models in diverse tasks such as image synthesis (Ramesh et al., 2021; Zhang et al., 2023a), video generation (Blattmann et al., 2023), and 3D modeling (Tevet et al., 2022; Wu et al., 2023). Tan et al. (2023); Zhong et al. (2023) generate dynamic traffic scenes based on user-specified descriptions expressed in natural language.

**Prompting.** Some works use language as an interface to interact with models, enabling users to request assistance or information. This includes obtaining explanations of underlying reasoning, and human-centric descriptions of model behavior (Brown et al., 2020; Sanh et al., 2022). Kuo et al. (2022) generate linguistic descriptions of predicted trajectories during decoding, capturing essential information about future maneuvers and interactions. More recent works employ large language models (LLMs) to analyze driving environments in a human-like manner, providing explanations of driving actions and the underlying reasoning (Xu et al., 2024; Fu et al., 2024; Sima et al., 2024; Wayve Technologies Ltd., 2023). This offers a human-centric description of the driving environment and the model's decision-making capabilities.

**Enriching.** Another line of work leverages LLMs' generalization abilities to enrich context embeddings, providing additional information for better prediction and planning (Guan et al., 2023). Zheng et al. (2024) integrate the enriched context information of LLMs into motion forecasting models. Wang et al. (2023b) use LLMs for data augmentation to improve out-of-distribution generalization. Others use pre-trained LLMs for better generalization during decision-making (Mao et al., 2024; Wen et al., 2024; Shao et al., 2024).

**Instructing.** Natural language can be used to issue explicit commands for specific tasks, distinct from conditioning (Ouyang et al., 2022; Brooks et al., 2023). The main challenge is connecting the abstractions and generality of language with environment-grounded actions (Raad et al., 2024). Shridhar et al. (2021) enable robotic control through language-based instruction. Zitkovich et al. (2023) incorporate web knowledge, enriching vision-language-action models for more generalized task performance. Huang et al. (2024) demonstrate the use of instructions to guide task execution in self-driving, with experiments in simulation environments.

Although these works align learned text representations with embeddings of other modalities, in contrast to our work, they do not measure the functional importance of features. To our knowledge, no prior work has explored the mechanistic interpretability of transformers in robotics applications.

## A.3 PARAMETERS FOR CLASSIFYING MOTION FEATURES

We classify motion trajectories with a sum less than $15°$ degrees as `straight`. When the cumulative angle exceeds this threshold, a positive value indicates `right` direction, while a negative value – exceeding the threshold in absolute terms – indicates a `left` direction. We classify speeds between $25 \, \mathrm{km\,h^{-1}}$ and $50 \, \mathrm{km\,h^{-1}}$ as moderate, speeds above this range as `high`, those below as `low`, and negative speeds as `backwards`. For acceleration, we classify trajectories as `decelerating`, if the integral of speed over time to projected displacement with initial speed is less than 0.9 times. If this ratio is greater than 1.1 times, we classify them as `accelerating`. For all other values, we classify the trajectories as having `constant` speed. We determine all threshold values by analyzing the distribution of the dataset.

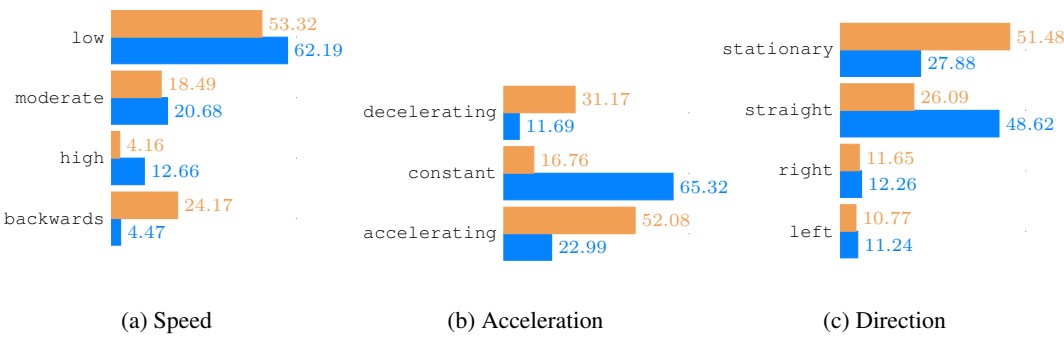

(a) Speed          (b) Acceleration          (c) Direction

Figure 8: Distributions of our motion features for the Argoverse 2 Forecasting (abbr. *AV2F*) and the Waymo Open Motion (abbr. *Waymo*) datasets. All numbers are percentages.

Figure 8 presents the distribution of motion subclasses across the datasets. Both datasets predominantly capture low-speed scenarios, with 62% of Waymo instances and 53% of AV2F instances falling into this category. Furthermore, a notable difference lies in the proportion of stationary vehicles, with AV2F exhibiting a significantly higher percentage (51%) compared to Waymo (28%). The Waymo dataset predominantly features vehicles with constant acceleration (65%) and traveling straight (49%), while the AV2F dataset has a higher proportion of accelerating instances (52%). Additionally, AV2F has a much larger proportion of instances involving backward motion (24%) compared to Waymo (4%). This disparity in motion characteristics highlights that the two datasets capture different driving environments and scenarios, with Waymo potentially focusing on highway or structured urban driving, while AV2F contains more diverse traffic situations.

## A.4 META-ARCHITECTURE OF MULTIMODAL MOTION TRANSFORMERS

We study multimodal motion transformers (Nayakanti et al., 2023; Wagner et al., 2024; Zhang et al., 2023b), which process motion, lane and traffic light data. The meta-architecture of these models is shown in Figure 9. These models generate motion $M_i$, map $K_j$, and traffic light $T_k$ embeddings using MLPs. Modality-specific encoders aggregate information from multiple embeddings with attention mechanisms (e.g., across multiple past timesteps for motion embeddings). Afterwards, in the motion decoder, learned motion queries $Q$ (i.e., a form of learned anchors) cross-attend to $M$, $K$, and $T$. Finally, an MLP projects the last hidden state of $Q$ into multiple motion forecasts, which are represented as 2D Gaussians for future positions in bird's-eye-view, along with their associated confidences. The differences between the models lie in the type of attention and fusion mechanisms they employ, as well as the used reference frames.

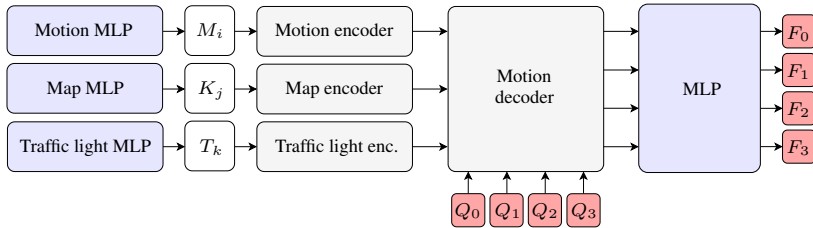

Figure 9: Motion transformer meta architecture of RedMotion, Wayformer, and HPTR.

## A.5 EARLY, HIERARCHICAL AND LATE FUSION IN MOTION ENCODERS

Fusion types for motion transformers are defined based on the information they process in the first attention layers. In early fusion, the first attention layers process motion data of the modeled agent, other agents, and environment context. In hierarchical fusion, they process motion data of the modeled agent, and other agents. In late fusion, they exclusively process motion data of the modeled agent.

A.6 FEATURE CORRELATION

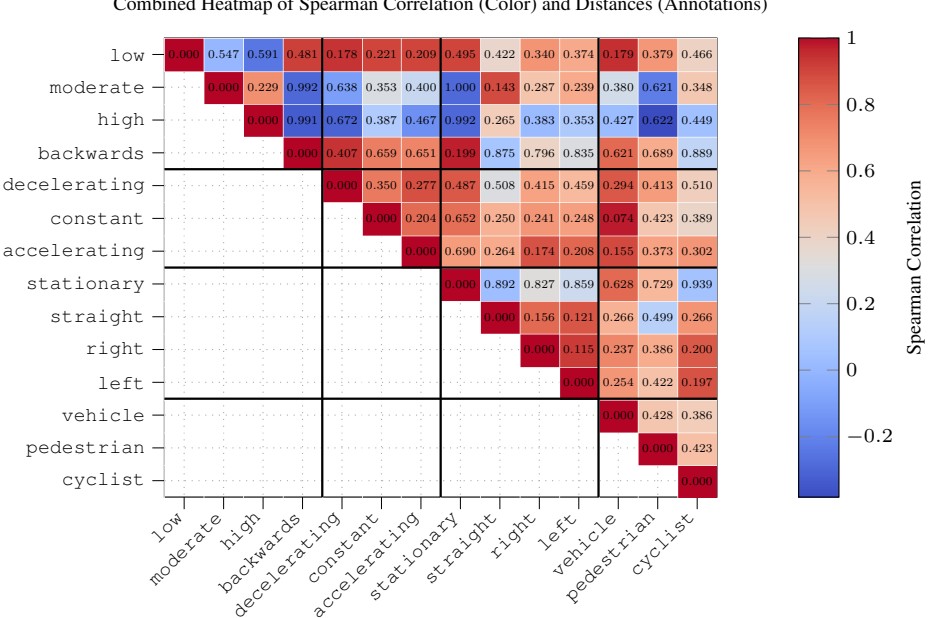

Figure 10: Heatmap representing Spearman correlation between feature cluster means for the Waymo Open Motion dataset. The values in the matrix indicate pairwise distances between clusters, normalized by the largest distance.

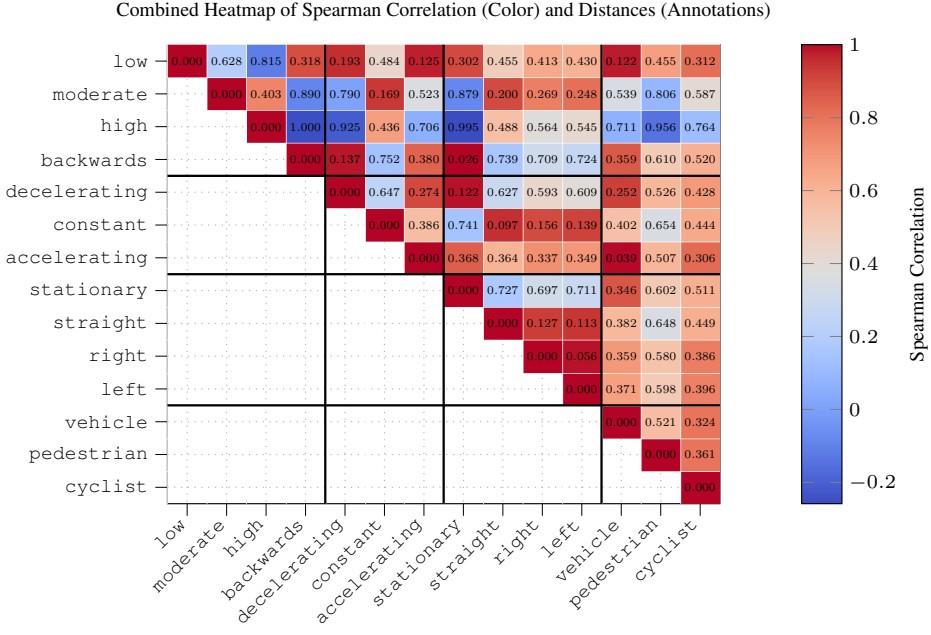

Figure 11: Heatmap representing Spearman correlation between feature cluster means for the Argoverse 2 Forecasting dataset. The values in the matrix indicate pairwise distances between clusters, normalized by the largest distance.

## A.7 EXPLAINED VARIANCE

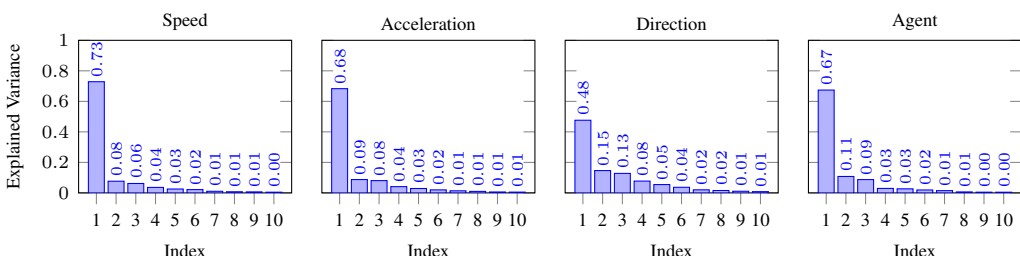

Figure 12: Explained variance for SAE across hidden latent dimensions 512, 256, 128, 64, 32, 16.

Figure 13: Explained variance for Plain-PCA.

## A.8 ADDITIONAL QUALITATIVE RESULTS

Figure 14 shows a qualitative example for our direction control vector from the Argoverse 2 Forecasting dataset. The left control leads to accelerated future motion, which is consistent with the common driving style of slowing down in front of a curve and accelerating again when exiting the curve. A strong right control makes the focal agent stationary. We hypothesize that it cancels out the actually driven left turn, resulting in a virtually stationary past.

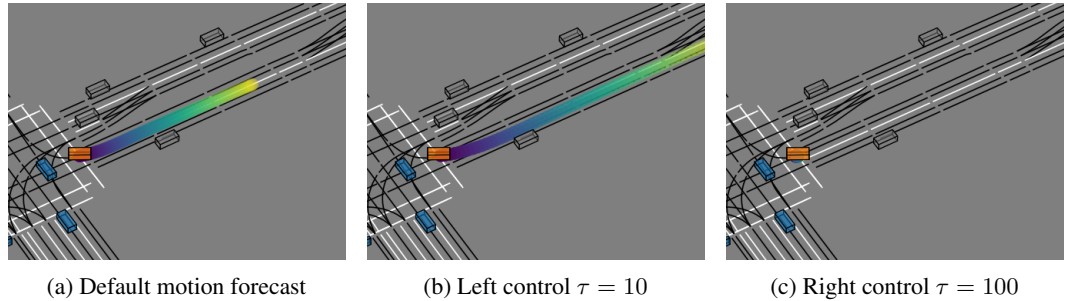

(a) Default motion forecast       (b) Left control $\tau = 10$       (c) Right control $\tau = 100$

Figure 14: **Modifying hidden states to control a left turning vehicle.** In subfigure (b) and (c), we apply our left-direction control vector and right-direction control vector. The focal agent is highlighted in orange, dynamic agents are blue, and static agents are grey. Lanes are black lines and road markings are white lines.

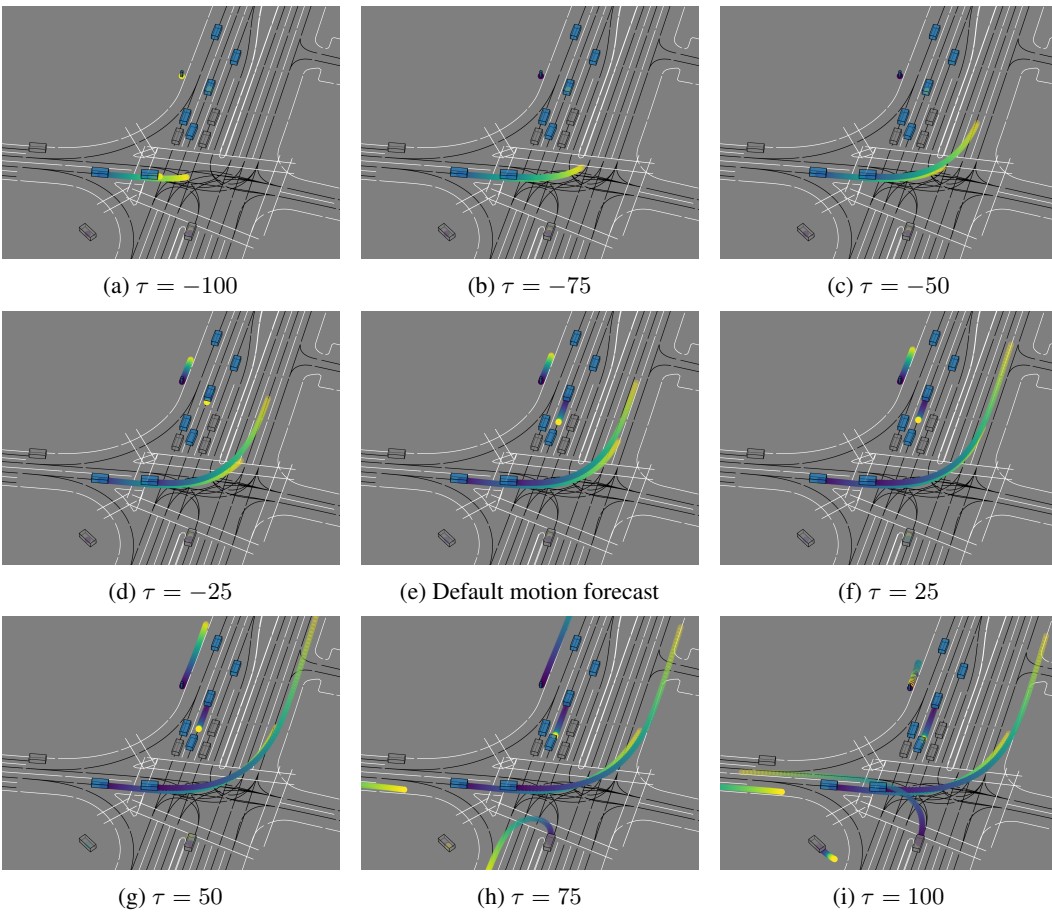

(a) $\tau = -100$       (b) $\tau = -75$       (c) $\tau = -50$

(d) $\tau = -25$       (e) Default motion forecast       (f) $\tau = 25$

(g) $\tau = 50$       (h) $\tau = 75$       (i) $\tau = 100$

Figure 15: Control vectors applied to all agents, showing consistent multi-agent behavior.

## A.9 COMPARISON OF CONTROL VECTORS USING PLAIN PCA AND SAE ACROSS VARIOUS SPARSE INTERMEDIATE DIMENSIONS

Table 6: Comparison of control vectors obtained with and without SAEs across sparse intermediate dimensions (512, 256, 128, 64, 32, 16). The tables on the left represent the pairwise angular distances between control vectors of the same SAE model, whereas those on the right represent the angular distances between control vectors of SAE and those derived from plain PCA. The control vector with a sparse intermediate dimension of 128 achieves the highest overall similarity.

| SAE-512 & SAE-512 | speed | acceleration | direction | agent |
|---|---|---|---|---|
| speed | 0.0 | 10.2 | 121.8 | 7.6 |
| acceleration | | 0.0 | 123.7 | 7.6 |
| direction | | | 0.0 | 126.9 |
| agent | | | | 0.0 |

| Plain PCA & SAE-512 | speed | acceleration | direction | agent |
|---|---|---|---|---|
| speed | 20.7 | 28.6 | 123.8 | 23.4 |
| acceleration | 19.1 | 23.0 | 128.5 | 18.6 |
| direction | 115.9 | 116.6 | 13.7 | 120.8 |
| agent | 19.4 | 24.4 | 130.2 | 18.3 |

| SAE-256 & SAE-256 | speed | acceleration | direction | agent |
|---|---|---|---|---|
| speed | 0.0 | 9.9 | 120.9 | 7.9 |
| acceleration | | 0.0 | 123.7 | 7.2 |
| direction | | | 0.0 | 126.3 |
| agent | | | | 0.0 |

| Plain PCA & SAE-256 | speed | acceleration | direction | agent |
|---|---|---|---|---|
| speed | 21.5 | 26.8 | 123.8 | 23.3 |
| acceleration | 20.3 | 21.0 | 128.7 | 18.7 |
| direction | 114.7 | 116.9 | 13.7 | 120.1 |
| agent | 20.8 | 23.1 | 130.2 | 18.7 |

| SAE-128 & SAE-128 | speed | acceleration | direction | agent |
|---|---|---|---|---|
| speed | 0.0 | 9.5 | 120.6 | 7.8 |
| acceleration | | 0.0 | 122.9 | 7.0 |
| direction | | | 0.0 | 125.8 |
| agent | | | | 0.0 |

| Plain PCA & SAE-128 | speed | acceleration | direction | agent |
|---|---|---|---|---|
| speed | 19.7 | 25.3 | 124.3 | 21.6 |
| acceleration | 19.2 | 20.0 | 128.8 | 17.5 |
| direction | 115.2 | 117.1 | 12.1 | 120.5 |
| agent | 19.5 | 21.8 | 130.4 | 17.1 |

| SAE-64 & SAE-64 | speed | acceleration | direction | agent |
|---|---|---|---|---|
| speed | 0.0 | 9.7 | 121.0 | 8.0 |
| acceleration | | 0.0 | 123.2 | 7.5 |
| direction | | | 0.0 | 126.3 |
| agent | | | | 0.0 |

| Plain PCA & SAE-64 | speed | acceleration | direction | agent |
|---|---|---|---|---|
| speed | 18.1 | 23.7 | 124.7 | 19.3 |
| acceleration | 19.3 | 19.9 | 128.9 | 16.5 |
| direction | 115.0 | 116.6 | 13.3 | 120.5 |
| agent | 19.8 | 21.9 | 130.5 | 16.4 |

| SAE-32 & SAE-32 | speed | acceleration | direction | agent |
|---|---|---|---|---|
| speed | 0.0 | 9.8 | 120.3 | 8.3 |
| acceleration | | 0.0 | 122.8 | 7.0 |
| direction | | | 0.0 | 125.8 |
| agent | | | | 0.0 |

| Plain PCA & SAE-32 | speed | acceleration | direction | agent |
|---|---|---|---|---|
| speed | 14.7 | 18.8 | 126.4 | 15.5 |
| acceleration | 18.0 | 15.5 | 130.3 | 14.1 |
| direction | 114.4 | 116.9 | 10.9 | 120.2 |
| agent | 18.1 | 17.6 | 132.0 | 13.4 |

| SAE-16 & SAE-16 | speed | acceleration | direction | agent |
|---|---|---|---|---|
| speed | 0.0 | 9.5 | 124.1 | 9.3 |
| acceleration | | 0.0 | 125.2 | 7.5 |
| direction | | | 0.0 | 129.3 |
| agent | | | | 0.0 |

| Plain PCA & SAE-16 | speed | acceleration | direction | agent |
|---|---|---|---|---|
| speed | 23.5 | 25.1 | 126.6 | 21.8 |
| acceleration | 28.4 | 26.0 | 128.9 | 23.5 |
| direction | 110.2 | 111.9 | 24.6 | 116.6 |
| agent | 28.0 | 26.8 | 131.0 | 22.5 |

## A.10 LOSS METRICS FOR SAES

We report the results for the epoch with the lowest total loss = $\ell_2$-loss $+ 3 \times 10^{-4} \cdot \ell_1$-loss. Note that the $\ell_2$ reconstruction loss is computed as the average of all partial losses for all embeddings, while the $\ell_1$ sparsity loss is computed as the sum of all partial losses.

Table 7: Loss metrics for SAEs across sparse intermediate dimensions, trained for 10.000 epochs.

| Dim | Best epoch | Total loss | $\ell_2$ reconstruction loss | $\ell_1$ sparsity loss |
|---|---|---|---|---|
| 512 | 9805 | 4.01 | 1.52 | 8270.70 |
| 256 | 9845 | 3.72 | 1.38 | 7823.98 |
| 128 | 9820 | 4.14 | 1.56 | 8608.95 |
| 64 | 9348 | 4.56 | 1.89 | 8894.97 |
| 32 | 9864 | 7.14 | 3.90 | 10 795.54 |
| 16 | 9956 | 17.44 | 13.37 | 13 576.57 |

## A.11 INFERENCE LATENCY

Table 8 shows inference latency measurements of a RedMotion model on the Waymo Open Motion dataset with and without activation steering with our control vectors. Our activation steering adds only about $1 \, \mathrm{ms}$ to the total inference latency. Since most datasets are recorded at $10 \, \mathrm{Hz}$ (e.g., Wilson et al. (2023); Ettinger et al. (2021)), it is common to define the threshold for real-time capability of self-driving stacks as $\leq 100 \, \mathrm{ms}$ . Considering the inference latency of recent 3D perception models (e.g., approx. $40 \, \mathrm{ms}$ for Wang et al. (2023a)), which must be called before motion forecasting, activation steering should not add significantly to the forecasting latency.

Table 8: **Inference latency without and with activation steering with our control vectors.** We measure the inference latency on one A6000 GPU using the PyTorch Lightning profiler and plain eager execution. We report the mean of 1000 iterations per configuration for the `predict_step`, including pre- and post-processing.

| Activation steering | Focal agents | Inference latency |
|---|---|---|
| False | 8 | 50.21  ms |
| True | 8 | 51.08  ms |

## A.12 CONTROL VECTORS ACROSS MODULES IN SPARSE AUTOENCODERS

Table 9: **Generating control vectors for hidden states of different modules.** Control vectors for speed generated in earlier modules achieve lower linearity scores for activation steering. Linearity measures for controlling: Pearson correlation coefficient, coefficient of determination ($R^2$), and straightness index.

| Autoencoder | Module $m$ | Pearson | $R^2$ | S-idx |
|---|---|---|---|---|
| SAE-128 | 2 | **0.993** | **0.984** | **0.988** |
| SAE-128 | 1 | 0.992 | 0.980 | 0.987 |
| SAE-128 | 0 | 0.959 | 0.654 | 0.933 |

## A.13 SENSITIVITY ANALYSIS FOR VARIOUS SPEED THRESHOLDS

Table 10: **Generating speed control vectors with different thresholds for low and high speed.** Decreasing the threshold for high speed marginally improves linearity scores, while increasing the threshold for low speed significantly worsens the linearity scores.

| Autoencoder | Low speed | High speed | Pearson | $R^2$ | S-idx |
|---|---|---|---|---|---|
| SAE-128 | $< 25 \, \mathrm{km \, h^{-1}}$ | $> 50 \, \mathrm{km \, h^{-1}}$ | 0.993 | 0.984 | 0.988 |
| SAE-128 | $< 25 \, \mathrm{km \, h^{-1}}$ | $25 \, \mathrm{to} \, 50 \, \mathrm{km \, h^{-1}}$ | 0.994 | 0.987 | 0.989 |
| SAE-128 | $25 \, \mathrm{to} \, 50 \, \mathrm{km \, h^{-1}}$ | $> 50 \, \mathrm{km \, h^{-1}}$ | 0.355 | $-0.734$ | 0.533 |

## A.14 CHOOSING A RANGE OF RELATIVE CHANGES IN FUTURE SPEED

Given the large range of scenarios in the Waymo dataset, we focus on relative speed changes within a range of $\pm 50\%$ to capture the most relevant speed variations (see Figure 3 in Ettinger et al. (2021)). Considering the approximated mean and standard deviation for each agent type (vehicles: $\mu \approx 12 \, \mathrm{m \, s^{-1}}$, $\sigma \approx 5 \, \mathrm{m \, s^{-1}}$, pedestrians: $\mu \approx 1.5 \, \mathrm{m \, s^{-1}}$, $\sigma \approx 0.7 \, \mathrm{m \, s^{-1}}$, and cyclists: $\mu \approx 7 \, \mathrm{m \, s^{-1}}$, $\sigma \approx 3 \, \mathrm{m \, s^{-1}}$) the $\pm 50\%$ range corresponds to speeds within approximately $\pm 1\sigma$ of the mean for each agent type.

### A.15 Evaluating a Koopman autoencoder

A consistent Koopman autoencoder (KoopmanAE) (Azencot et al., 2020) is a bidirectional method that models temporal dynamics between embeddings. The learned latent space approximates a Koopman-invariant space where dynamics evolve linearly. Adapted to the SAE configurations, we train an encoder and a decoder with one layer each and a latent dimension of 128. We use learned linear projections to decode Koopman operator approximations $C, D \in \mathbb{R}^{128 \times 128}$ from intermediate representations. For the first 10 time steps, we encode the embedding and predict the next embedding using $C$, while for the last 10 time steps, we encode the embedding and predict the previous embedding using $D$.

Afterwards, we use the KoopmanAE instead of an SAE to fit control vectors (see Section 3.3). Figure 16 shows the calibration curve for the resulting speed control vector. The range of $\tau$ values is approximately $100\times$ smaller than for the SAE-based control vector shown in Figure 7b.

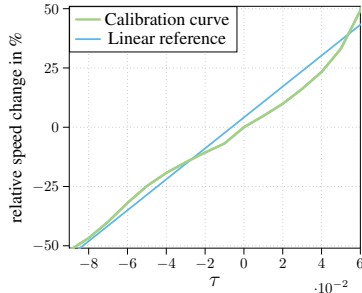

Figure 16: **Calibration curve for a speed control vector optimized using the KoopmanAE.** The range of $\tau$ values is significantly lower than for plain PCA and SAE-based control vectors (cf. Figure 7), yielding lower $R^2$ scores as shown in Table 3.

### A.16 Plain PCA-based speed control vector for the AV2F dataset

Figure 17 shows the calibration curve for a plain PCA-based speed control vector for a RedMotion model trained on the AV2F dataset (cf. Section 5.4). To suppress the effects of different trajectory lengths, we trained this model on a configuration of the AV2F dataset with the same trajectory lengths as in the Waymo dataset. In contrast to the control vectors for the Waymo dataset, this control vector cannot reduce the speed by more than 3%. Therefore, we center the range of $\tau$ values instead of the range of relative speed changes to compare this control vector with those for the Waymo dataset.

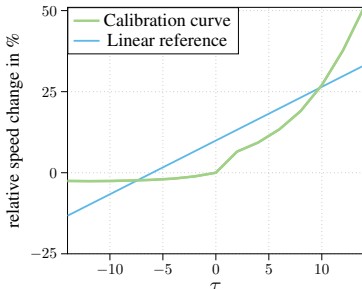

Figure 17: **Calibration curve for a plain PCA-based speed control vector for the AV2F dataset.** In contrast to the control vectors for the Waymo dataset, this control vector cannot reduce the speed by more than 3%.

Moreover, we used the low and moderate speed classes to fit this control vector, as the low and high speed classes did not yield good results. We hypothesize that this is due to the different distributions of the datasets shown in Figure 8.

### A.17 JumpReLU compensates activation shrinkage in ConvSAEs

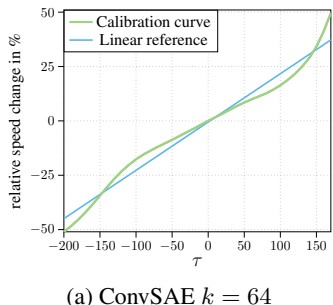
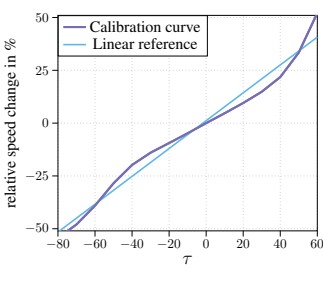

(a) ConvSAE $k = 64$          (b) ConvSAE $k = 64$ JumpReLU

Figure 18: JumpReLU compensates activation shrinkage as reflected in a smaller range of $\tau$ values for the same range of relative speed changes.

The range of temperatures is much higher for the ConvSAE than for the JumpReLU version of this sparse autoencoder (ConvSAE $k = 64$ JumpReLU). We hypothesize that this is due to activation shrinkage (Rajamanoharan et al., 2024b). Therefore, the JumpReLU configuration of this SAE-type leads to a significantly smaller $\tau$ range, which in turn leads to higher $R^2$ scores (see Table 3).

### A.18 Motion forecasting metrics

Following Wilson et al. (2023); Ettinger et al. (2021), we use the average displacement error (minADE), the final displacement error (minFDE), and their respective Brier variants, which account for the predicted confidences. Furthermore, we compute the miss rate, and overlap rate to evaluate motion forecasts. All metrics are computed using the minimum mode. Accordingly, the metrics for the trajectory closest to the ground truth are measured.

### A.19 Neural collapse

Neural collapse metrics capture structural patterns in feature representations, focusing on clustering, geometry, and alignment. Class-distance normalized variance (CDNV), also referred to as "$\mathcal{NC}1$", quantifies the degree to which features form class-wise clusters by measuring the variance within feature clusters of each class $c$ relative to the distances between class means. CDNV provides a robust alternative to methods that compare between- and within-cluster variation for assessing feature separability (Galanti et al., 2022).

$$\mathcal{NC}1_{c,c'}^{\text{CDNV}} = \frac{\sigma_c^2 + \sigma_{c'}^2}{2\|\mu_c - \mu_c'\|_2^2}, \quad \forall c \neq c'$$

