# OpenReview forum: "Words in Motion: Extracting Interpretable Control Vectors for Motion Transformers"
_ICLR.cc/2025/Conference — ICLR 2025 Poster_

### Official Review · Reviewer_5JLb · 2024-10-20

**Soundness:** 2
**Presentation:** 2
**Contribution:** 2
**Rating:** 5
**Confidence:** 4

**Summary:**

This paper aims to interpret hidden states and control them at inference. It assesses whether human-interpretable features are embedded in hidden states of motion transformers using neural collapse. The latent space properties are used to fit control vectors for each interpretable feature. The control vectors are optimized using sparse autoencoding and enforcing sparsity results in a more linear relationship between control vector temperatures and forecasts.

**Strengths:**

1. The paper introduces an approach for interpreting hidden states in motion forecasting models by leveraging neural collapse and linear probing.

2. The paper uses sparse autoencoders to optimize control vectors and enhance the linearity between control temperatures and forecasts.

3. Applying the method to self-driving cars and motion prediction shows its relevance to real-world applications.

**Weaknesses:**

1. The related work section does not clearly and effectively highlight the similarities and differences between prior research and your work.

2. The study lacks baseline comparisons. Are there existing methods that could be used to evaluate your proposed approach?

3. The paper lacks novelty, as it does not introduce linear probes or control vectors but simply applies these existing methods to motion control.

**Questions:**

1. Will using a sparse autoencoder significantly increase computational costs?

2. This paper is not well written. Some of the design motivations are not clearly explained.

---

> ### Author Response · Authors · 2024-11-21
>
> Thank you for your feedback on our work.  Below are our responses to your specific questions and the weaknesses you identified.
>
> > **W1:** The related work section does not clearly and effectively highlight the similarities and differences between prior research and your work.
>
> We have added a paragraph on how our work differs from existing works (see lines 125-131):
>
> Our method differs from prior works in several aspects. We measure neural collapse in multimodal models for motion forecasting (i.e., regression) instead of unimodal vision classifiers (Papyan et al., 2020) or language models (Wu & Papyan, 2024). Rather than steering hidden state changes across all modules (i.e., neural trajectories) as in Zou et al. (2023), we use only the hidden states in the last module of the motion encoder. Furthermore, we do not use our sparse autoencoders during inference as in Cunningham et al. (2023), but to optimize control vectors beforehand, resulting in negligible computational overhead (see Appendix A.11).
>
>
> > **W2:** The study lacks baseline comparisons.
>
> To the best of our knowledge, no prior work has explored the mechanistic interpretability of transformers in robotics applications.
> However, we significantly extended our experiments to ablate different activation functions, and layer types. Overall, the regular SAEs achieve the highest Pearson and R\textsuperscript{2} scores. JumpReLU activation functions improve the R\textsuperscript{2} scores marginally compared to ReLU activation functions. Please see Table 2 in the revised version.
>
>
> > **W3:** The paper lacks novelty.
>
> To the best of our knowledge, we are the first to to improve control vectors for activation steering with sparse autoencoders and to connect neural collapse (representation learning theory) and control vectors (mechanistic interpretability).
>
>
> > **Q1:** Will using a sparse autoencoder significantly increase computational costs?
>
> We do not use our sparse autoencoders during inference as in Cunningham et al. (2023), but to optimize control vectors beforehand, resulting in negligible computational overhead. Additionally we measure the inference latency with and without our activation steering (see appendix A.11).
>
>
> > **Q2:** This paper is not well written. Some of the design motivations are not clearly explained.
>
> We revised Figure 1 and Section 3.3 to better highlight our design motivations. Furthermore, we provided the details on our feature and parameter choices in Appendix A.1, A.4, A.8 in the initial version.

---

> > ### Comment · Reviewer_5JLb · 2024-11-25
> >
> > Thank you for the author's response. I still have a few concerns regarding your work.
> >
> > 1. Although you added a paragraph on how your work differs from existing work, it is still unclear why your method is preferable or what specific advantages it offers over existing approaches.
> >
> > 2.  You mentioned that no prior work has explored the mechanistic interpretability of transformers in robotics applications. Are there any other methods for interpreting features in hidden states that could be transferred to robotics applications for performance comparison?

---

> > > ### Author Response · Authors · 2024-11-25
> > >
> > > We agree that we had to transfer methods for interpreting hidden states to our application for performance comparison. Below we describe how we did this and the specific advantages over existing approaches:
> > >
> > > Our method enables using control vectors for hidden states of motion data by quantizing motion features into discrete classes (the method of Zou et al. (2023) requires discrete positive and negative examples).
> > >
> > > Our method optimizes activation steering with control vectors in two ways:
> > > 1. We generate more distinct control vectors by using the sparse intermediate representations of SAEs, resulting in a more linear relationship between control temperatures and future speeds. For comparison, the first row of Table 2 and Figure 7 (a) show the results of vanilla control vectors (Zou et al. 2023) within our framework.
> > > 2. We lower the computational overhead by controlling only the last hidden state of the motion encoder rather than all hidden states and by using the sparse autoencoders beforehand rather than at inference (see previous response or lines 129 - 131 and Appendix A.11).
> > >
> > > Furthermore, we ablate different SAEs within our framework (see Table 2, row 2: Cunningham et al. (2023), row 3: Rajamanoharan et al. (2024b)) and further variants with different layer types (MLPMixer and convolutional), patch/ kernel sizes and activation functions.

---

> > > > ### Comment · Reviewer_5JLb · 2024-12-02
> > > >
> > > > Thank you for the author's response. However, I think the paper still requires further interpretations and baseline comparisons. Therefore, I would like to maintain my original score.

---

> > > > > ### Comment · Reviewer_FS1W · 2024-12-02
> > > > >
> > > > > As another reviewer who has no horse in the race here, I find this justification of maintaining the score of 5 unclear. They performed more baseline experiments and adjusted the writing as you requested. I find it quite well written and their rebuttal clear.

---

> > > > > > ### Comment · Reviewer_6xxM · 2024-12-02
> > > > > >
> > > > > > As another reviewer who also has no horse in the race here, I find this championing of the paper unnecessary, regardless of the original reviewer's comment. The work as it currently stands still requires significant reworking, including a proper contextualization and description of the motion transformer in the main article and implications of results from comparison for different steering approaches to be of interest to interpretability and the broader ICLR community.

---

> > > > > > > ### Comment · Reviewer_FS1W · 2024-12-02
> > > > > > >
> > > > > > > :D no worries 6xxM, I am simply commenting that it is unclear what is still missing in this paper given the revisions - I am not "championing" the paper, I am engaging in discourse that is completely standard while reviewing manuscripts; I would expect other reviewers to engage with constructive reviews, hence my asking for clarification of why this paper is below par (score 5). If your opinion is that is just requires editing the text of the manuscript (not new experiments), to me that is above the bar. I assume authors in good faith would change this before the camera ready.

---

> > > > > > > > ### Comment · Reviewer_6xxM · 2024-12-02
> > > > > > > >
> > > > > > > > Ah okay! no worries then! I'll also be giving the full paper another careful read before the reviewing period ends to see if I can increase my score.

---

### Official Review · Reviewer_QCbU · 2024-11-05

**Soundness:** 2
**Presentation:** 2
**Contribution:** 2
**Rating:** 3
**Confidence:** 3

**Summary:**

The paper introduces a method for extracting interpretable representations in transformer-based motion forecasting models: (i) leverage neural collapse to identify a structured latent space where features form interpretable clusters, (ii) use sparse autoencoders to optimize "control vectors" for each interpretable feature. This method is evaluated on three motion forecasting models, demonstrating effective feature manipulation and zero-shot generalization.

**Strengths:**

- The paper addresses the under-explored area of mechanistic interpretability in motion forecasting, providing a timely study at this intersection.
- The paper presents thorough quantitative and qualitative experiments, demonstrating the effectiveness of the interpretability method across different models and scenarios.

**Weaknesses:**

- While the paper is new in connecting mechanistic interpretability with motion forecasting, the technical components (neural collapse and sparse autoencoders) have been studied in LLMs. The claim of `a significant step towards mechanistic interpretability and controllability of transformer models` (L452) seems quite overstated. Further clarification on the unique technical contributions would strengthen the paper.
- The paper extends interpretability techniques from LLMs to the motion domain; however, the contexts differ fundamentally: LLMs deal with discrete tokens, while motion forecasting operates in a continuous space. The proposed method quantizes continuous variables like speed and acceleration into discrete categories, which might oversimplify the difference. A deeper discussion on the impact of this discretization and the influence of specific parameters (e.g., in L160) would enhance understanding of the challenges and significance of applying mechanistic interpretability to motion forecasting.

**Questions:**

- How sensitive are the results to the thresholds mentioned in Appendix A.3?

---

> ### Author Response · Authors · 2024-11-21
>
> Thank you for your feedback on our work.  Below are our responses to your specific questions and the weaknesses you identified.
>
> > **W1a:** While the paper is new in connecting mechanistic interpretability with motion forecasting, the technical components (neural collapse and sparse autoencoders) have been studied in LLMs. (...) Further clarification on the unique technical contributions would strengthen the paper.
>
> To the best of our knowledge, we are the first to to improve control vectors for activation steering with sparse autoencoders and to connect neural collapse and control vectors.
>
> > **W1b:** The claim of a significant step towards mechanistic interpretability and controllability of transformer models (L452) seems quite overstated.
>
> We agree that our experiment target motion transformers. Therefore, we rephrased the "transformer models" to "motion transformers", as in the rest of the paper.
>
> > **W2:** The paper extends interpretability techniques from LLMs to the motion domain; however, the contexts differ fundamentally: LLMs deal with discrete tokens, while motion forecasting operates in a continuous space. The proposed method quantizes continuous variables like speed and acceleration into discrete categories, which might oversimplify the difference.
>
> Although we use discrete classes of motion features to fit control vectors, our temperature-scaled activation steering enables fine-grained control of future speeds (see calibration curves in Figure 7).
>
> > **Q1:** How sensitive are the results to the thresholds mentioned in Appendix A.3?
>
> We additionally ablated speed thresholds (see lines 466-469 and appendix A.15)
> Our method is more sensitive to low than to high speed, likely because most pedestrians in the dataset are moving with low speeds.

---

> > ### Comment · Reviewer_QCbU · 2024-11-27
> >
> > Thank you for your response.
> >
> > After carefully reading your response and the comments from the other reviewers, I think the primary concern from my initial review -- `overstated claim` -- still stands.
> >
> > The paper is positioned as a novel interpretability method but limits its evaluation to motion transformers.
> > If the goal is to present a general method, it needs to be supported with experiments in other general setups. Alternatively, the paper could narrow its scope, but be more specific on the unique challenges and contributions within interpretable robotics.
> >
> > The current gap between the claim and supporting evidence seems to have caused confusion for other reviewers as well:
> >
> > - Reviewer 6xxM: `this is a bold claim`, `justify the claim`
> > - Reviewer 5JLb: `lacks novelty` `baseline comparisons`
> > - Reviewer KZpZ: `explained as an application area of their method, not the main focus`
> >
> > Overall, I think the paper takes on an interesting direction and would benefit from substantial revisions.

---

> > > ### Author Response · Authors · 2024-11-27
> > >
> > > We kindly invite you to review our responses and the changes marked in blue in our revised paper, as your response still seems to be based on your initial impression.
> > >
> > > Nevertheless, we further clarify our responses to your comments and reaffirm the scope and contributions of our work:
> > >
> > > 1. Title and scope (overstated claim):
> > >
> > >     * The title explicitly focuses on motion transformers and reflects our narrowed scope. Repeated use of the word "motion" emphasizes our focus on this domain and avoids overstating generality.
> > >
> > >     * We revised the conclusion as well
> > >        * see line 500, and our responses to you, 5JLb, 6xxM, and KZpZ,
> > >
> > >       and added addidtional appendices on self-driving-specific details
> > >        * see A.17, response to 6xxM, on the meta-architecture of motion transformers
> > >        * see A.11, responses to 6xxM, FS1W, 5JLb, on inference latency measurements
> > >        * see A.16, response to 6xxM, on choosing a range of changes in future speeds
> > >        * see A.15, responses to you, 6xxM, KZpZ on a sensitivity analysis w.r.t. speed thresholds.
> > >
> > > 2. Novelty, unique contributions, and baseline comparison:
> > >
> > >    * We are the first to to improve control vectors for activation steering with sparse autoencoders and to connect neural collapse and control vectors (see lines 125-131, repsonses to you, 6xxM, 5JLb).
> > >
> > >    * We present a detailed analysis of control vector linearity with respect to their temperatures, using this as a metric to compare different SAE configurations -- an analysis that has not been conducted in prior work (see Table 2, responses to 6xxM, FS1W, 5JLb, KZpZ).
> > >
> > >    * We evaluate our approach on three unique motion transformers with commonly used trajectory datasets (as mentioned as the 2nd strength in your review).
> > >
> > > We trust that our revisions have addressed your concerns and hope our clarifications allow for a fair assessment of our work.

---

> ### Comment · Reviewer_QCbU · 2024-11-27
>
> Thank you for the detailed response to my earlier comments. However, I must emphasize that my previous response was **not** `based on initial impression`.
>
> You argue that the `repeated use of the word "motion" emphasizes our focus on this domain and avoids overstating generality`. Frankly, I find this argument misleading.
>
> The storyline of the paper places limited emphasis on motion, even in its latest version. For example
> - the abstract sets the stage as `transformer-based models`, `interpret these hidden`, `control them at inference`
> - the 1st paragraph in intro: `interpretability is crucial for **many real-world applications**`
> - the 2nd paragraph in intro: `deep learning models`, `representations of transformer-based models`
> - the 3rd paragraph in intro: `our method allows for controlling transformer-based forecasting models`
> - motion is only mentioned in the last paragraph, towards the end of the intro section
> - similarly, in the related work section spanning over 60 lines, motion is mentioned only briefly (4 lines) at the end.
>
> As stated previously, I believe the paper would benefit from `substantial revisions` and believe this is a `fair assessment`.
>
> Please let me know if there are aspects I may have misunderstood.

---

> > ### Author Response · Authors · 2024-11-27
> >
> > We regret that our revisions and responses have not convinced you.
> >
> > Please note that:
> > * Beyond the title, we explicitly highlight our focus on motion forecasting already in the second sentence of the abstract.
> > * The introduction begins with a general perspective before narrowing to our application, with the final paragraph dedicated to it.
> > * In the related work section, we clearly state that no comparable works exist on our application topic.
> >
> > We thank you for taking the time to engage in these discussions.

---

### Official Review · Reviewer_KZpZ · 2024-11-09

**Soundness:** 2
**Presentation:** 1
**Contribution:** 2
**Rating:** 3
**Confidence:** 5

**Summary:**

The authors have proposed finding an interpretable feature in the motion transformer and then enabling the control of the prediction based on the found interpretable features. To this end, authors claim that the interpretable feature can be found with the linear probing the hidden state, then use a control vector combined with sparse autoencoder to control the output of the motion transformer.

**Strengths:**

The problem is well-motivated: building an interpretable and controllable motion prediction network.

**Weaknesses:**

1. I think the overall writing needs to be improved in multiple aspects.
- First, it is hard to understand why the author has focused on motion transformers in the introduction. After reading section 3 (method part), I can understand why the author needs an interpretable method for motion transformers, but in the introduction, it is explained as an application area of their method, not the main focus.
- While the author suggests they have used neural collapse to measure the human interpretable feature, it is not known what neural collapse is in the introduction.

2. Neural collapse is the most important term in the paper, but it is not defined clearly (or clearly mathematically), so it is unable to understand what the authors are using. Note that the authors have argued that they have used neural collapse as a metric, so we need a mathematical way to measure it. I think the author should define the following in a clear (or mathematical) way.
- What is "Neural collapse"?
-  What is an "interpretable feature"?
- What is "motion features"?

3. While the author mentioned they have used sparse autoencoder (SAE), it is not clear which one they have used [1,2,3].
- Which layer did the author train the SAE? SAE is usually trained on the hidden state of the transformer [1,2,3], and the layer is always selected carefully.
- Can the author report that the SAE is trained well? For instance, the reconstruction error of the SAE. (I think the author mentioned it is reported in the Appendix, but could not find it).

4. Can the author explain the goal of the motion transformer? What is the input of the motion transformer and output of the motion transformer?

5. It is not clear why the author used SAE for the collability. There are several activation steering methods [4,5]. I think it is great to claim the benefit of using SAE over other methods.

Overall, I think the paper needs to improve in terms of writing, especially the introduction and the method part. Moreover, the author could improve the overall experimental details and analysis mentioned above.

Reference\
[1] Scaling and evaluating sparse autoencoders\
[2] Jumping Ahead: Improving Reconstruction Fidelity with JumpReLU Sparse Autoencoders\
[3] Improving Dictionary Learning with Gated Sparse Autoencoders\
[4] In-context Vectors: Making In Context Learning More Effective and Controllable Through Latent Space Steering\
[5] In-Context Learning Creates Task Vectors

**Questions:**

See the weakness above.

---

> ### Author Response · Authors · 2024-11-21
>
> Thank you for your feedback on our work.  Below are our responses to your specific questions and the weaknesses you identified.
>
>
> > **W1a:** After reading section 3 (method part), I can understand why the author needs an interpretable method for motion transformers, but in the introduction, it is explained as an application area of their method, not the main focus.
>
> This is a slight misunderstanding, we mention the specific models we analyze in detail in the introduction (see lines 46 - 51) and do not mention other applications of our method.
>
> > **W1b:** While the author suggests they have used neural collapse to measure the human interpretable feature, it is not known what neural collapse is in the introduction.
>
> In the revised paper, we explicitly mention the term neural collapse in the introduction, as well (see line 33).
>
> > **W2a:** Neural collapse is the most important term in the paper, but is not defined clearly (or clearly mathematically), so it is unable to understand what the authors are using.
>
> While neural collapse is an important concept in our paper, we consider control vectors and sparse autoencoders for activation steering to be more important for our work.
> Please refer to section 2.2 paragraph 2 for details on neural collapse. In the revised version, we added the formula for the Class Distance Normalized Variance, which we use in addition to linear probing accuracy to measure neural collapse, to the appendix (see lines 1040 - 1049).
>
> > **W2b:** What is an "interpretable feature"? What is "motion features"?
>
> Please refer to section 3.1.
>
> > **W3a:** While the author mentioned they have used sparse autoencoder (SAE), it is not clear which one they have used.
>
> We use the sparse autoencoder introduced in Cunningham et al. (2023) and have added the reference to line 198. In the revised paper, we also include an ablation for other sparse autoencoders with different layer types and activation functions (see lines 428 - 437 and table 2).
>
> > **W3b:** Which layer did the author train the SAE?
>
> We trained the SAE on the hidden states from Module 2 of the motion encoder (see Section 4.3). In the revised version, we ablate hidden states from earlier modules as well (see lines 466 - 469 and appendix A.14).
>
>
> > **W3c:** Can the author report that the SAE is trained well? For instance, the reconstruction error of the SAE. (I think the author mentioned it is reported in the Appendix, but could not find it).
>
> See appendix A.1 table 5.
>
> > **W4:** Can the author explain the goal of the motion transformer? What is the input of the motion transformer and output of the motion transformer?
>
> Motion forecasting, please refer to the introduction (lines 46 - 51).
>
> > **W5:** It is not clear why the author used SAE for the collability.
>
> Our main contribution is that we optimize control vectors for activation steering using SAEs. Our motivation is that sparse intermediate representations of SAEs enable more linear decompostion of our interpretable features, and hence, more distinct control vectors (see section 3.3, lines 198 - 206).

---

> > ### Comment · Reviewer_KZpZ · 2024-11-28
> > **Thank you for your response**
> >
> > Thank you for your time and effort for the rebuttal. However, I still think the paper should be improved. First, I still think the paper needs to be written more clearly. I have read the paper again, but it was hard to understand, and there are multiple missing definitions. For instance, I was not asking about the informal goal of the motion transformer (which is very intuitive), rather I was asking about the formal definition of input, output. Without such definitions, the reader can misunderstand the paper. In the same sense as above, I was not asking about the section where the "interpretable feature" or "motion feature" was defined (rather asking for a formal definition). Second, I also quite agree with reviewer 5JLb on why this paper offers specific advantages over existing approaches (as I have mentioned in the original review). While the claim is that SAE offers more interpretable features, it is hard to understand it without a formal definition of interpretable features (or evaluations that measure this). Finally, I also agree with the reviewer QCbU that the paper needs substantial revisions. In this regard, I am maintaining my original score. I thank the authors again for the time and effort for the rebuttal.

---

> > > ### Author Response · Authors · 2024-11-29
> > >
> > > Thank you for taking the time to revisit our paper and the other reviews.
> > >
> > > In the revised version, Appendix A.12 provides a detailed description of the inputs and outputs of the motion transformers used in our work, which are available on GitHub. Similarly, in the initial submission version, we defined "motion features" and analyzed their correlations. Because these features are physically measurable (i.e. acceleration, velocity, direction, and agent type), we use them as "interpretable features", as described in Section 3.3.
> > >
> > > We appreciate your engagement in these discussions.

---

### Official Review · Reviewer_FS1W · 2024-11-09

**Soundness:** 3
**Presentation:** 3
**Contribution:** 3
**Rating:** 8
**Confidence:** 3

**Summary:**

The paper introduces a method to interpret and control transformer-based motion forecasting models by analyzing hidden states through the lens of neural collapse. Using linear probes and control vectors, it maps hidden states to interpretable features, such as speed and direction, enabling insight into the model’s decision-making and the ability to manipulate forecasts without retraining. This approach enhances model interpretability, though its reliance on neural collapse and associated computational demands present some (minor) challenges, and testing other base models beyond SAE vs. PCA would be interesting.

**Strengths:**

- **Interpretability:** The method takes motion transformers, and maps hidden states to human-interpretable features, thus clarifying the model’s decision-making process. In general, interpretability is an important area.

- **Controllability:** Control vectors allow manipulation of specific motion features (e.g., speed, acceleration) at inference time without retraining, enabling intuitive model adjustments.

- **Zero-shot Generalization:** The interpretable control vectors support generalization to unseen scenarios, like different driving styles or environments, compensating for domain shifts in the data.

- **Nice integration across disciplines:** I like using neuro-inspired linear probing and their quantization method using natural language. I think this shows an elegant combination of techniques that can be leveraged to build interpretable systems.

**Weaknesses:**

- **Reliance on Neural Collapse:** The method's effectiveness depends on well-defined hidden state clusters. If neural collapse is weak, the extracted features and control vectors may be less reliable.

- **Limited Feature Scope:** The approach primarily focuses on basic motion features (e.g., speed, acceleration, direction) and could be expanded to capture more complex motion patterns and interactions with the environment.

- **Limited Baselines**: Currently, they primarily focus on comparisons for the control vectors are to PCA, but do not consider methods that inherently have dynamics, such as temporal convolutional networks (TCNs), nonlinear embedding methods that consider time-series inputs (one prominent example would be Schneider et al. 2023 Nature), or RNNs. While not critical for their argument, they should minimally discuss this in a limitation/discussion section.

- **Computational Overhead:** Using sparse autoencoders increases computational demands, potentially extending training time and resource requirements; it would be ideal to estimate the computational resources used and those needed to utilize their code and models that they note will be released.

**Questions:**

See weaknesses, above. I would be happy for the authors to provide commentary on the baselines and computational resources, then I would be happy to raise my score.

---

> ### Author Response · Authors · 2024-11-21
>
> Thank you for your feedback on our work.  Below are our responses to your specific questions and the weaknesses you identified.
>
> > **W1:** If neural collapse is weak, the extracted features and control vectors may be less reliable.
>
> Recent literature suggests that neural collapse is a universal phenomenon that occurs in the terminal stages of successful training. Training with different models (Papyan et al. 2020; Wu & Papyan, 2024), objectives (Andriopoulos et al., 2024), and learning paradigms (Ben-Schaul et al. 2023).
>
>
> > **W2:** The approach focuses on basic motion features and could be expanded to capture more complex motion patterns and interactions with the environment.
>
> We agree and plan to evaluate this in future work.
>
> > **W3:** Limited baselines.
>
> We significantly extended our experiments to ablate different activation functions, and layer types. Overall, the regular SAEs achieve the highest Pearson and R\textsuperscript{2} scores. JumpReLU activation functions improve the R\textsuperscript{2} scores marginally compared to ReLU activation functions. Please see Table 2 in the revised version. Furthermore, we mentioned other embedding methods as future work (see line 512).
>
> > **W4:** Computational Overhead.
>
> We do not use our sparse autoencoders during inference as in Cunningham et al. (2023), but to optimize control vectors beforehand, resulting in negligible computational overhead. Additionally we measure the inference latency with and without our activation steering (see appendix A.11).

---

> > ### Comment · Reviewer_FS1W · 2024-11-21
> >
> > Thank you for your clarifications and additional experiments. The revisions are very much appreciated, and I have raised my score.

---

### Official Review · Reviewer_6xxM · 2024-11-10

**Soundness:** 2
**Presentation:** 3
**Contribution:** 2
**Rating:** 5
**Confidence:** 4

**Summary:**

The paper focuses on the interpretability and control of transformer-based motion forecasting models. The authors aim to interpret hidden states of motion transformers and control them during inference. They use neural collapse to assess whether human-interpretable features are embedded within hidden states. The authors fit control vectors for steering during inference for the interpretable features. They further finetune these vectors using sparse autoencoders. They show that enforcing sparsity leads to a more linear relationship between the strength of the steering and interpretable features. They apply this approach to various motion forecasting models with different fusion mechanisms and environment representations. Finally, they address domain shifts using the control vectors and enable zero-shot generalization.

**Strengths:**

1. The paper applies interpretability approaches to transformer-based models beyond the natural language domain and propose neural collapse as a metric of interpretability.

2. The authors use sparse autoencoder-based steering for improving control vector linearity for motion control and zero-shot generalization.

3. The authors have done significant work to apply the proposed method to multiple motion forecasting architectures and datasets.

**Weaknesses:**

1. The paper relies heavily on neural collapse as a measure for the model learning clusters of interpretable features. The authors should verify the feature clusters are indeed distinct by comparing the within-class and between-class variance.
2. The L1 sparsity in the training objective is known to induce feature shrinkage and result in poor reconstructions (Wright and Sharkey, 2024). The authors should compare l1, l2, and reconstruction loss for other SAE architectures that do not induce feature shrinkage such as TopK or JumpReLU SAEs, and use the best-performing architecture to finetune control vectors and show their impact on control linearity. If training new SAEs seem out-of-scope, then they should report the % loss recovered from the SAEs and ensure that they are in an acceptable range.
3. The authors should test their approach on other naturally occurring domain shifts like various traffic densities, scenes with different weather conditions, etc.
4. Details on the training of the sparse autoencoders such as choice of learning rate and schedule, batch size, number of epochs, etc. as missing.
5. Other minor comments:
           a. The authors give a brief explanation of neural collapse in paragraph 2 of the introduction but do not explicitly mention the term.
           b. Along similar lines, the authors use terms like domain shift and fusion mechanisms for the transformer architectures and zero-
               shot generation, they should be properly defined and contextualized for their specific problem.
           c. The reference for sparse autoencoders used in Section 4.4 is for the Gated SAEs paper, however, the authors do not use that
               specific architecture. The reference should be changed appropriately.

**Questions:**

1. In the introduction, the authors claim that they identify that interpretable features are embedded in hidden states of transformer-based models. Since this is a well-known observation, the reviewer is curious about how the work adds to the existing knowledge in the mechanistic interpretability literature.
2. In the conclusion, the authors claim that they take a significant step towards the mechanistic interpretability and controllability of the transformer models. This is a bold claim - how do the results from the paper justify the claim and add to the pre-existing knowledge in the broader transformer interpretability literature?
3. How do different feature quantization thresholds affect results? The rationale for these specific thresholds should be better justified by showing the dataset statistics. The authors mention the choice was based on insights from Seff et al., 2023 - what were the specific insights the authors used for their choice of classes?
4. In Section 3.2, the authors claim to use the mean of the standard deviation to measure collapse - is this correct or should this metric be just the standard deviation?`
5. For the control vectors found using pca, what is the variance explained in each case?
6. The implications of neural collapse on the post-training processes of a model like transferability and generalization is still an open question (Vignesh Kothapalli, 2024). Can the authors better justify the use of neural collapse to show the validity of their probe accuracies?
7. How would the control effects change when combining multiple control vectors (e.g., speed + direction)?
8. How do the control vectors found using one approach compare to the other, in terms of their cosine similarity?
9. What was the intuition behind choosing the temperature range of [-50,50]?
10. Can the authors comment on how to extend their method to continuous control features rather than discrete classes as they have used in the paper?
12. Will the authors release both code and pre-trained models, including the trained sparse autoencoders?

---

> ### Author Response · Authors · 2024-11-21
>
> Thank you for your feedback on our work.  Below are our responses to the weaknesses you identified and your specific questions.
>
> > **W1:** The authors should verify the feature clusters are indeed distinct by comparing the within-class and between-class variance.
>
> We measured them and added them in Section 5.1, Lines 320-322. Consistent with the CDNV value, these variances are higher for the Argoverse2 dataset than for the Waymo Open Motion dataset. We hypothesize this is due to longer past motion sequence, allowing for a greater range of potential movements.
>
> > **W2:** The authors should compare (...) other SAE architectures (...) such as TopK or JumpReLU SAEs, and use the best-performing architecture to finetune control vectors and show their impact on control linearity.
>
> We extended our ablations by evaluating different layer types and activation functions including convolutional layers, MLPMixer layers, and JumpReLU. The results are shown in Table 2. Overall, the regular SAEs achieve the highest Pearson and R\textsuperscript{2} scores.
>
> The ConvSAE with a kernel size of 64 achieves the highest straightness index, yet the lowest R\textsuperscript{2} scores. We hypothesize that this is due to feature shrinkage (Rajamanoharan et al., 2024). Therefore, the JumpReLU configuration of this SAE-type leads to a significantly smaller $\tau$ range (see Figure 15, page 22), which in turn leads to higher R\textsuperscript{2} scores (see Table 2, page 9). The regular SAEs do seem to not exhibit such feature shrinkage in our application (please refer to R\textsuperscript{2} scores in Table 2).
>
> > **W3:** The authors should test their approach on other naturally occurring domain shifts like various traffic densities, scenes with different weather conditions, etc.
>
> While we recognize the importance of evaluating robustness under diverse scenarios, our current study is based on trajectory datasets, which do not provide the ability to modify or control conditions like weather or traffic density.
>
> > **W4:** Details on the training of the sparse autoencoders such as choice of learning rate and schedule, batch size, number of epochs, etc. as missing.
>
> Details on the training of sparse autoencoders are provided in Section 4.4 (see lines 252-258 in the initial version or 267-271 in the revised version).
>
>
> > **W5a:** The authors do not explicitly mention the term neural collapse in paragraph 2.
>
> We explicitly mentioned neural collapse (see line 33).
>
>
> > **W5b:** Define domain shift, fusion mechanisms and zero-shot generalization
>
> We define a domain shift in motion forecasting as less aggressive diving styles, which result in lower future speeds (see lines 428-430 in the initial version and 474-476 in the revised version). Fusion mechanisms in motion transformers (early, hierarchical and late) are defined in appendix A.4 in the initial version and A.5 in the revised version.

---

> > ### Author Response · Authors · 2024-11-21
> >
> > > **Q1:** The reviewer is curious about how the work adds to the existing knowledge in the mechanistic interpretability literature.
> > We have added a paragraph on how our work differs from existing works (see lines 125-131):
> >
> > Our method differs from prior works in several aspects. We measure neural collapse in multimodal models for motion forecasting (i.e., regression) instead of unimodal vision classifiers (Papyan et al., 2020) or language models (Wu & Papyan, 2024). Rather than steering hidden state changes across all modules (i.e., neural trajectories) as in Zou et al. (2023), we use only the hidden states in the last module of the motion encoder. Furthermore, we do not use our sparse autoencoders during inference (Cunningham et al., 2023), but to optimize control vectors beforehand, resulting in negligible computational overhead.
> >
> >
> >
> > > **Q2:** In the conclusion, the authors claim that they take a significant step towards the mechanistic interpretability and controllability of the transformer models. This is a bold claim.
> >
> > We agree that our experiment target motion transformers. Therefore, we rephrased the "transformer models" to "motion transformers", as in the rest of the paper.
> >
> >
> > > **Q3:** How do different feature quantization thresholds affect results?
> >
> > We additionally ablated speed thresholds (see lines 466-469 and appendix A.15). Our method is more sensitive to low than to high speed, likely because most pedestrians in the dataset are moving with low speeds.
> >
> >
> > > **Q4:** The authors claim to use the mean of the standard deviation to measure collapse.
> >
> > This is a slight misunderstanding. We use the standard deviation of the l2 normalized embeddings to measure representation collapse, not neural collapse. Representation collapse describes that learned embeddings (across all classes/ features) collapse to redundant or trivial solutions (see lines 174-175 in the initial version and 182-183).
> >
> >
> > > **Q5:** For the control vectors found using pca, what is the variance explained in each case?
> >
> > We added the explained variance for our SAE-based sparse intermediate representations and for hidden states (plain PCA) to the apppendix A.9.  As in Zou et al. (2023) the first principal corresponds to a significantly higher explained variance than the following ones.
> >
> >
> > > **Q6:** Can the authors better justify the use of neural collapse to show the validity of their probe accuracies?
> >
> > We follow Ben-Schaul et al. (2023) and use linear probes to measure neural collapse. They show that the accuracy of linear probes is consistent with the accuracy of nearest class center classifiers, which are typically used to measure neural collapse (see footnote 3 in our paper).

---

> ### Author Response · Authors · 2024-11-21
>
> > **Q7**:  How would the control effects change when combining multiple control vectors?
>
> Thanks for the remark. We expect our method to work for the other processed modalities such as lanes and traffic light states as well (see meta architecture of motion transformers in Appendix A.12). We plan evaluate this in future work.
>
> > **Q8**: How do the control vectors found using one approach compare to the other, in terms of their cosine similarity?
>
> Please see appendix A.1.
>
>
> > **Q9**:  What was the intuition behind choosing the temperature range of [-50,50]?
>
> We acknowledge that fixing temperature range instead of speed range is less intuitive. Therefore, we revised this. In the revised version we fixed the speed range to ±50%. Please see Table 2 and Appendix A.16.
>
>
> > **Q10**: Can the authors comment on how to extend their method to continuous control features rather than discrete classes.
>
> Although we use discrete classes of motion features to fit control vectors, our temperature-scaled activation steering enables fine-grained control of future speeds (see calibration curves in Figure 7).
>
>
> > **Q11**: Will the authors release both code and pre-trained models, including the trained sparse autoencoders?
>
> We will release both code and pre-trained models.

---

> > ### Comment · Reviewer_6xxM · 2024-11-27
> >
> > Thank you for providing additional experiments and updates to the paper. After looking at the additional details provided and other reviewer's comments, I regret to inform the authors that I would be unable to increase the scores.

---

> > > ### Author Response · Authors · 2024-11-28
> > >
> > > Thank you for your time.
> > >
> > > Given that we tried to address all your concerns, we'd love to hear in which direction we should improve our paper so that it better aligns with your expectations.

---

### Meta-Review · Area_Chair_B6dY · 2024-12-24

**Metareview:**

The problem studied in this submission is considered interesting by the majority of reviewers, attracting favourable remarks about clear motivation and broader interest in studying interpretability techniques for transformers beyond the language domain. In addition, other positive comments included the ability to manipulate specific motion features (e.g., speed, acceleration) at inference time without retraining) as well as the application to multiple motion forecasting architectures and datasets.

On the flip side, several reviewers criticised the submission for requiring major revision in terms of clarity of writing while also requesting authors to address a perceived lack of baseline comparisons. While the reviewers included an ablation study testing various SAE alternatives, this does not go far enough. Other criticisms revolve around the novelty of the approach (beyond changing the application domain) and justification for claims made throughout the manuscript. Finally, the method is considered to rely heavily on neural collapse, for which reviewers may prefer stronger robustness guarantees.

On balance, this submission is unfortunately not yet ready for publication at ICLR.

**Additional Comments On Reviewer Discussion:**

Positive discussion, both between authors and reviewers and among reviewers themselves.

---

### Decision · Program_Chairs · 2025-01-22

Accept (Poster)